



**Source apportionment of ambient particle number concentrations in central Los**
**Angeles using positive matrix factorization (PMF)**
M.H. Sowlat[1], S. Hasheminassab[1], C. Sioutas[1]
[1] University of Southern California, Department of Civil and Environmental Engineering
*Corresponding author
3620 S. Vermont Ave. KAP210, Los Angeles, CA 90089
E-mail: sioutas@usc.edu
Telephone: 213-740-6134
Fax: 213-744-1426
**Abstract**
In this study, the Positive Matrix Factorization (PMF) receptor model (version 5.0)
was used to identify and quantify major sources contributing to particulate matter (PM)
number concentrations, using PM number size distributions in the range of 13 nm to 10 µm
combined with several auxiliary variables, including black carbon (BC), elemental and
organic carbon (EC/OC), PM mass concentrations, gaseous pollutants, meteorological, and
traffic counts data, collected for about 9 months between August 2014 and 2015 in central
Los Angeles, CA. Several parameters, including particle number and volume size distribution
profiles, profiles of auxiliary variables, contributions of different factors in different seasons
to the total number concentrations, diurnal variations of each of the resolved factors in the
cold and warm phases, weekday/weekend analysis for each of the resolved factors, and
correlation between auxiliary variables and the relative contribution of each of the resolved
factors, were used to identify PM sources. A six-factor solution was identified as the
optimum for the aforementioned input data. The resolved factors comprised nucleation,
traffic 1, traffic 2 (having a larger mode diameter than traffic 1 factor), urban background
aerosol, secondary aerosol, and soil/road dust. Traffic sources (1 and 2) were the major
contributor to PM number concentrations, collectively making up to above 60% (60.8-68.4%)
of the total number concentrations during the study period. Their contribution was also
significantly higher in the cold phase compared to the warm phase. Nucleation was another
major factor significantly contributing to the total number concentrations (an overall



contribution of 17%, ranging from 11.7% to 24%), having a larger contribution during the
warm phase than in the cold phase. The other identified factors were urban background
aerosol, secondary aerosol, and soil/road dust, with relative contributions of approximately
12% (7.4-17.1), 2.1% (1.5-2.5%), and 1.1% (0.2-6.3%), respectively, overall accounting for
about 15% (15.2-19.8%) of PM number concentrations. As expected, PM number
concentrations were dominated by factors with smaller mode diameters, such as traffic and
nucleation. On the other hand, PM volume and mass concentrations in the study area were
mostly affected by sources having larger mode diameters, including secondary aerosols and
soil/road dust. Results from the present study can be used as input parameters in future
epidemiological studies to link PM sources to adverse health effects as well as by policy
makers to set targeted and more protective emission standards for PM.
**1. Introduction**
Numerous epidemiological studies have provided compelling evidence linking exposure
to ambient particulate matter (PM) with increased risk of respiratory and cardiovascular
diseases, hospitalization, and premature mortality (Brunekreef and Forsberg, 2005;Dockery
and Stone, 2007;Miller et al., 2007;Pope et al., 2004;Pope Iii et al., 2002;Gauderman et al.,
2015). According to the most recent global burden of disease study, over 3 million premature
deaths annually occur all around the globe due to exposure to ambient PM (Lim et al., 2013).
It should, however, be noted that most of these epidemiological studies have related the
aforementioned health outcomes with solely the mass concentrations of PM and, therefore, do
not adequately represent submicron particles (Ogulei et al., 2007), mainly because this PM
fraction contributes negligibly to total ambient PM mass (Delfino et al., 2005;Vu et al.,
2015). More recently, studies have associated human health effects with particles
characteristics other than mass concentration, including size, number concentration, chemical
composition, and even surface area (Brook et al., 2010;Kasumba et al., 2009;Lighty et al.,
2000;Chen et al., 1991;Dreher et al., 1997;Oberdörster et al., 1994;Peters et al., 1997;Delfino
et al., 2010;Davis et al., 2013). Even though our knowledge of which particle characteristic
(mass, size, surface area, etc.) can be considered as the best predictor for human health
outcomes is limited, there is growing evidence highlighting the critical role of particle size
and number concentrations from a human health effect perspective (Vu et al., 2015). For
example, studies have indicated that ultrafine particles (UFP, i.e. particles with an
aerodynamic diameter of <100 nm) have higher toxicity per unit mass (Donaldson et al.,
1998;Li et al., 2003;Nel et al., 2006;Oberdörster et al., 2002), have higher deposition



efficiencies in the lung (Venkataraman, 1999), and penetrate deeper into the alveolar regions
of lungs (Sioutas et al., 2005). Additionally, several studies have also found that PM number
concentrations (mostly UFPs) can be associated with adverse effects on human health,
particularly for cardiovascular diseases (Delfino et al., 2005;Peters et al., 1997;Wichmann et
al., 2000).
As a consequence, regulations on PM number concentrations have already been
implemented on motor vehicle emissions in few countries. For instance, the Euro 5b and 6
have set a limit to particle number emission factors, in addition to particle mass emission
limits, for heavy-duty and gasoline vehicles (http://www.dieselnet.com/standards/eu/ld.php,
accessed 20 October 2015). It is also expected that this approach be gradually adopted in
other parts of the world (Friend et al., 2013), based mainly on the critical health implications
of the PM number concentrations, especially in smaller fractions like UFP. This emphasizes
the necessity of identification and quantification of PM sources based on number as well as
mass (Friend et al., 2013). This allows for source-specific assessment of health effects of
exposure to PM to provide us with the knowledge required to develop efficient control
strategies for PM emissions from major sources to minimize those health effects (Yue et al.,

17  2008).

Positive Matrix Factorization (PMF) is one of the most widely used receptor models
that have been successfully applied to identify and quantify sources of atmospheric particles.
The vast majority of previous efforts has been devoted to the identification of sources that
contribute to the mass of particles using PMF on chemically-speciated PM mass data in
different parts of the world (Sowlat et al., 2013;Sowlat et al., 2012;Dutton et al., 2010;Lim et
al., 2010;Alleman et al., 2010;Sofowote et al., 2015). Recently, attempts have been made to
characterize sources that contribute to particle number, rather than mass, using PMF applied
to particle number size distribution data. These studies have adopted different approaches in
their source apportionment, including: (1) using particle number size distribution together
with gaseous pollutants, chemical composition, meteorological, or traffic data in the PMF
analysis (Beddows et al., 2015;Harrison et al., 2011;Kasumba et al., 2009;Ogulei et al.,
2007;Ogulei et al., 2006b;Thimmaiah et al., 2009;Zhou et al., 2005), (2) using particle
number size distribution and chemical composition data in separate and/or combined PMF
runs (Beddows et al., 2015;Gu et al., 2011), (3) comparing PMF results with actual events
during the study period (Ogulei et al., 2007), and (4) simply correlating the PMF results with
gaseous pollutants data (Friend et al., 2013;Friend et al., 2012;Kim et al., 2004). It is





noteworthy that the major factors resolved by these studies have been nucleation, traffic,
secondary aerosol, urban background, and wood burning.
Numerous studies have been performed in Los Angeles evaluating PM number
concentrations as well as size distributions, with a focus on vehicular emissions as a major
source of particle number in urban areas (Singh et al., 2006;Zhang et al., 2005). Source
apportionment of atmospheric particles has also been extensively studied in Los Angeles, but
almost all of the studies have focused on the contribution of different sources to PM mass
rather than PM number concentration (Ham and Kleeman, 2011;Hasheminassab et al.,
2013;Hwang and Hopke, 2006;Kim and Hopke, 2007;Kim et al., 2010;Schauer and Cass,
2000). To the best of our knowledge, no source apportionment studies have ever been
performed in Los Angeles on particle number size distributions using PMF. The only study
providing a source apportionment of particle number concentrations in Los Angeles is that of
Brines et al. (2015), in which major sources contributing to particle number concentrations
were identified in five high-insolation cities around the world (Barcelona, Madrid, Roma, Los
Angeles, and Brisbane) using the k-means clustering method. It should, however, be noted
that, in case of Los Angeles, the Brines et al. (2015) study used particle number size
distribution data for a rather limited time period (i.e., 3 months); moreover, the studied size
distributions ranged from 13 nm to 400 nm, thus excluding potentially important PM sources
contributing to the larger size fractions of $PM_{2.5}$ and/or $PM_{10}$.
In the present work, we collected high-resolution (5-min measurements), wide-spectrum
particle number size distribution data (i.e., 13 nm to 10 μm, covering the nucleation, Aitken,
accumulation, and coarse PM modes) over a long period of time (i.e., 9 months, covering
both warm and cold seasons) in a location near downtown of Los Angeles, California, to
identify and quantify sources contributing to particle number concentrations using the most
recent version of the PMF model (version 5.0). We also included gaseous pollutants (i.e., CO,
NO, $NO_2$, $O_3$), particle mass ($PM_{10-2.5}$ and $PM_{2.5}$), meteorological (temperature, relative
humidity (RH), and wind speed), black carbon (BC), elemental carbon (EC) as well as
primary (POC) and secondary organic carbon (SOC), and traffic (counts of light-duty
vehicles (LDVs) and heavy-duty vehicles (HDVs)) data as inputs to help identify the factors
resolved by the model. Results from the present study can be used as a platform for future
health effect studies to estimate the source-specific impact of exposure to PM from a number
concentration perspective, which is critical for development and establishment of abatement
strategies and standards in order to minimize the most relevant health outcomes.



**2. Methodology**
*2.1. Sampling site*
Continuous measurements were carried out at the particle instrumentation unit (PIU)
located on the University of Southern California's (USC) park campus, approximately 3 km
south of downtown Los Angeles, CA. The PIU is located within approximately 150 m
downwind of a routinely congested interstate freeway, i.e. I-110, and is also in close
proximity to parking and construction facilities. Previous studies conducted by this research
group have indicated that the PIU is a mixed urban site that is also heavily impacted by
vehicular emissions (Geller et al., 2004;Moore et al., 2007;Hasheminassab et al., 2014b).
*2.2. Sampling time, method, and instrumentation*
Continuous measurements were conducted at the PIU from August 2014 through March
2015 as well as in August 2015. To obtain number size distribution of atmospheric particles
in the size range of 14 -760 nm (mobility diameter), a Scanning Mobility Particle Sizer
(SMPS$^{TM}$, TSI Model 3081) was used, which was connected to a Condensation Particle
Counter (CPC, model 3020, TSI Inc., USA). Particles in the size range of 0.3-10 μm (optical
diameter) were measured using an Optical Particle Sizer (OPS$^{TM}$, Model 3330, TSI Inc.,
USA). The time resolution for these two instruments was 5 min. The OPS instrument was
calibrated by the manufacturer using Polystyrene Latex (PSL) particles, which have a
dynamic shape factor of 1 (i.e., spherical particles) and a refractive index of 1.59. It should
also be noted that the measurements provided by the OPS instrument depend primarily on the
refractive index and the dynamic shape factor (Hasheminassab et al., 2014b). Numerous
studies have indicated that for spherical particles, the size selection offered by optical particle
counters, such as the OPS instrument, is quite similar to the actual physical diameter of the
particle being measured (Chen et al., 2011;Hasheminassab et al., 2014b;Hering and
McMurry, 1991;Reid et al., 1994). That said, there is compelling evidence in the literature
supporting the fact that the refractive index and the dynamic shape factor for ambient
aerosols in urban areas (such as Los Angeles) are quite similar to those of PSL
particles (Covert et al., 1990;Ebert et al., 2004;Hänel, 1968;Kent et al., 1983;Stolzenburg et
al., 1998;Strawa et al., 2006;Watson et al., 2002). To further evaluate this assumption, we
used the Multi-Instrument Manager (MIM$^{TM}$) software, developed by TSI Inc., USA, which
estimates the refractive index and dynamic shape factor of aerosols from parallel SMPS and
OPS scans. The output from this software indicated that the average real part of the refractive
index for the aerosols collected in this study was 1.59±0.01 and their dynamic shape actor





was 0.99±0.02. This finding is also in concert with the results of Hasheminassab et al.
(2014b), which reported an average shape factor of near unity at the same sampling site,
using the apparent and material density of aerosols. Hence, further adjustment of OPS sizing
was deemed unnecessary and the OPS size distribution, with the original size selection, was
merged with the SMPS size spectra. More detailed information on the sensitivity of the OPS
sizing to the refractive index and the dynamic shape factor of aerosols can be found in
(Hasheminassab et al., 2014b).

9         Black carbon (BC) concentrations, with a time resolution of 15 min, were measured

using a portable Aethalometer (Magee scientific, model AE-42). Hourly concentrations of
elemental carbon (EC) and organic carbon (OC) were measured using a semi-continuous
EC/OC carbon aerosol analyzer (Model 4, Sunset Laboratory Inc., USA), using the
thermal/optical transmittance measurement protocol of the National Institute of Occupational
Safety and Health (NIOSH 5040). By applying the "EC tracer method", Saffari et al. (2016)
estimated the primary organic carbon (POC) and secondary organic carbon (SOC)
concentrations from total OC at the same location. These two parameters (i.e., POC and
SOC) were also used as input parameters in the PMF model, as they can provide valuable
input regarding the detection of primary and secondary sources of PM. The EC tracer method
has been discussed in detail elsewhere (Day et al., 2015;Saffari et al., 2016). Briefly, the main
assumption in this method is that EC and POC are released from similar sources; therefore,
this approach is most applicable where combustion is the main source of ambient POC (Day
et al., 2015). It is noteworthy that, in the present study, the sampling site was located in close
proximity to a major freeway, thereby making the EC tracer suitable for the data collected in
this location, as it has also been used in previous studies in the same sampling site (Polidori et
al., 2007;Saffari et al. 2016) as well as similar locations in the Los Angeles basin (Na et al.,
2004; Strader et al., 1999). In this method, the following equations can be used after
determining the ratio of POC to EC to estimate the concentration of SOC:
$$POC = [OC/EC]_p \times EC + b \qquad (1)$$
$$SOC = OC - POC \qquad (2)$$
where, $[OC/EC]_p$ is the POC to EC ration; b is the intercept of the linear regression between
POC and EC, which is considered to be the portion of POC associated with non-combustion
emissions. It is also noteworthy that we used the "high EC edge method" to determine





observations with a high probability of dominant POC contribution, which is believed to be a
more accurate method for the identification of the $[OC/EC]_p$ ratio compared to the traditional
approach, as discussed by Day et al. (2015), and has also been successfully applied in a
number of previous studies (Harrison and Yin, 2008; Lim and Turpin, 2002; Na et al. 2004).
*2.3. Auxiliary variables*

7       To help better identify the factors resolved by the PMF model, additional parameters,

including gaseous pollutants (i.e., CO, NO, $NO_2$, and $O_3$) and particulate matter mass
concentrations in two size fractions (i.e., $PM_{10-2.5}$ and $PM_{2.5}$), meteorological parameters (i.e.,
temperature, relative humidity, and wind speed), and traffic flow data (counts of LDVs and
HDVs), were also included in the model as auxiliary variables. Hourly concentrations of
particulate mass and gaseous pollutants together with hourly measurements of meteorological
parameters were acquired from the online data base of California Air Resources Board
(CARB), for the sampling site located in Downtown Los Angeles (North Main St.),
approximately 3 km to the northeast of the PIU. The hourly traffic flow data were acquired
from the nearest vehicle detection station (VDS) to our sampling site on the Freeway I-110,
operated by the freeway performance measurement system (PeMS), under the
California Department of Transportation (CalTrans). Table 1 provides a summary statistics of
the input parameters to the PMF model in this study. To achieve the same time resolution
across all variables, we calculated hourly-averaged data points for all variables.
*2.4. Meteorology in central Los Angeles*

23       To evaluate the impact of meteorological conditions on factor contributions as well as

to better identify the resolved factors based on their expected seasonal trends, the study
period was partitioned into two phases, i.e., colder phase (from November to February) and
warmer phase (from August to October as well as March), and the model outputs, except for
factor profiles, are presented for each phase accordingly. Figure 1 illustrates the diurnal
variation of important meteorological parameters, namely, temperature, RH, wind speed, and
solar radiation, in the cold and warm phases. As can be seen from the figure, on average,
temperature was 5-7 °C higher in the warm phase than in the cold phase, although the trends
were similar in both phases. Minimum temperatures were observed in the early morning
(coinciding with morning rush hours), while maximum temperatures were seen at around
noon. Conversely, RH peaked at night and exhibited a minimum in the early afternoon. RH



was also slightly higher in the warm phase than in the cold phase. As expected, wind speed
peaked in the early afternoon during the warm phase and slightly shifted to the evening in the
cold phase, while the slowest winds were blown during nighttime. The wind speeds were also
higher in the warm phase compared to the cold phase. Solar radiation had a consistent trend
in both phases, peaking at noon, with the levels being higher in the warm phase than in the
cold phase, as one would expect. Similar trends and levels were also observed by
Hasheminassab et al. (2014b) in central LA, indicating the occurrence of stable atmospheric
conditions during nighttime until morning rush hours, especially in colder months of the year.
*2.5. PMF model*
PMF, first developed by Paatero and Tapper (1993), is a multivariate statistical model
used for identifying and quantifying the contribution of different sources to a set of samples
using the fingerprints of those sources. This multivariate factor analysis tool decomposes a
matrix of speciated data into two sub-matrices, i.e., factor profiles and source contributions,
as shown below (Krecl et al., 2008):
$X = G.F + E$ (3)
where, X is the matrix of samples (here, particle number size distribution together with
auxiliary variables data); G is the matrix containing source contributions; F is the matrix
containing factor profiles; and E is the residual matrix.
The above equation can also be expressed mathematically, as the following (Norris et
al., 2014):
$$x_{ij} = \sum_{k=1}^{p} g_{ik} f_{kj} + e_{ij}$$ (4)
where, $x_{ij}$ is the PM number concentration (or concentration of another auxiliary
species) for the *i*th sample and the *j*th size bin (or species); *p* is the number of factors that
contribute to the PM number concentrations; $g_{ik}$ is the relative contribution of *k*th factor to *i*th
sample; $f_{ik}$ is the PM number concentration of *j*th size bin in the *k*th factor; and $e_{ij}$ is the
residual (observed–estimated) value for the *i*th sample and *j*th size bin.
With the constraint that no sample can have a significantly negative contribution and
using a least-square method, the PMF then resolves factor profiles and contributions by
attempting to minimize the Q value, as shown below (Paatero, 1997;Paatero and Tapper,

31  1994):



$$Q = \sum_{i=1}^{n} \sum_{j=1}^{m} \left[ \frac{x_{ij} - \sum_{k=1}^{p} g_{ik} \cdot f_{kj}}{u_{ij}} \right]^2 \qquad (5)$$

where, $u_{ij}$ is the uncertainty associated with the sample $x_{ij}$.
One of the advantages of the PMF model is weighting every single value in the input
data matrix using user-provided uncertainties, enabling the model to allow for measurement
confidence in resolving the factor profiles and contributions (Paatero et al., 2014). In the
present work, since no measurement uncertainties were available for the input parameters, we
applied the method suggested by Ogulei et al. (2006a;2006b) and Zhou et al. (2014) to
calculate the uncertainties for individual data points inserted into the model. For this purpose,
measurement errors were first estimated for each data point using the following equation:
$\sigma_{ij} = C_1 (N_{ij} + \overline{N}_j)$      (6)
where, $\sigma_{ij}$ is the estimated measurement error for the $i$th sample and $j$th size bin (or
concentration of auxiliary variables); $C_1$ is an empirical constant usually between 0.01 and
0.05; $N_{ij}$ is the observed number concentration for the $i$th sample and $j$th size bin (or
concentration of auxiliary variables); and $\overline{N}_j$ is the arithmetic mean of the PM number
concentrations for the $j$th size bin (or concentration of auxiliary variables).
The value of the measurement method obtained from the above equation is then used to
calculate the measurement uncertainty, according to the following equation:
$S_{ij} = \sigma_{ij} + C_2 \max(|x_{ij}|, |y_{ij}|)$     (7)
where, $S_{ij}$ is the calculated uncertainty associated with the $i$th sample and $j$th size bin; $C_2$
is an empirical constant usually between 0.1 and 0.5; and $Y_{ij}$ is the value calculated by the
model for $x_{ij}$. In the present work, $C_1$ and $C_2$ values of 0.05 and 0.1 were chosen to obtain the
most physically interpretable solution using a trial and error approach.
In the present study, the most recent version of the PMF model, version 5.0, newly
released by the United States Environmental Protection Agency (PMF guide), was used.
Uncertainties associated with the resolved factor profiles were estimated using three error
estimation methods, namely, Displacement (DISP) analysis, Bootstraps (BS) method, and a
combination of DISP and BS methods (BS-DISP). For the DISP analysis, a solution was
considered valid if the observed drop in the Q value was below 0.1% and there was no factor
swaps for the smallest $dQ_{max}$ (i.e., 4). For the BS method, 100 runs were selected and a
solution was considered valid if all of the factors had a mapping of above 90%. For the BS-





DISP analysis, a solution was considered valid if the observed drop in the Q value was below
0.5% (Brown et al., 2015;Paatero et al., 2014).
The PMF model was run in the robust mode, which down-weights the effect of values
with high uncertainties (i.e., values set as "weak" in the model) on the final solution resolved
by the model (Brown et al., 2015). Missing values were replaced by interpolating the
previous and the next data points in the matrix; however, to decrease the effect of these
replaced values on the final solution, their uncertainty was set as three times the mean
uncertainty for that species (that is practically what the model does to set a species as
"weak"). Based on the recommendations presented by Brown et al. (2015), genuine zero
values were included in the input matrix. Particle number concentration (PNC) was selected
as the "total variable", and the PMF model automatically turned it into a weak species by
increasing its uncertainty by a factor of 3. An extra model uncertainty of 5% was also set to
account for errors that are not covered in the input uncertainty values (Reff et al., 2007), since
the uncertainty matrix only includes the effect of random as well as experimental errors.
*2.6. Input matrices*
The model was run in two different scenarios, one with EC/OC data, which included
1053 samples of 131 species, and one without EC/OC data, which included 2976 samples of
129 species. This was due mainly to the fact that the EC/OC data were being collected in
parallel for a different study that coincided with the current work in a span of time shorter
than the entire study period. Therefore, in order to keep the large number of samples from the
main study (i.e., 2976) as well as to use the critical advantage of having EC/OC data in the
factor identification process, it was decided to run the PMF model in two different scenarios,
one including EC/OC data and one without these data. It should also be noted that although
the latter matrix contained BC data, this variable was excluded from the former matrix to
avoid double counting, as EC was already included in the dataset. The results of the PMF run
including the EC/OC data are provided in the supplementary information (Figure S1).
**3. Results and discussion**
*3.1. Overview of the data*
Table 2 presents the statistical characteristics of the species included in the PMF model.
In this table, signal-to-noise (S/N) ratio is a parameter that indicates if the variability in the
measurements is real or within the data noise. In the current version of the model, i.e., PMF





5.0, the method used for calculating the S/N ratio has been updated compared to the previous
versions, resolving the disadvantages associated with the previous method of S/N calculation
(for a more detailed discussion on the S/N calculation methods, see SI). In the current
method, if the resulting S/N ratio is above 1, it can be concluded that the species has a
reliable signal. As reported in Table 2, all the species in the input matrix had S/N ratios well
above 1, indicating very strong signals for all the variables. Figure S2 also illustrates the
correlation between the measured and PMF-predicted total number concentrations for the
entire sampling period. As can be seen in the figure, the high correlation between measured
and predicted values ($R^2$=0.99) and very close to 1 slope of the regression line indicate that
the PMF model has been successful in modeling the input data and apportioning the total PM
number concentrations to the resolved factors.
Figure 2 depicts the average number and volume size distributions of all the input data
to the PMF model by phase, which were collected during the entire study period. As shown in
the figure, the vast majority of the particles were smaller than 100 nm, and the number
concentration had a mode diameter at around 40 nm. Additionally, a significantly higher
number concentration was observed in the cold phase compared to the warm phase, which is
consistent with the results from the previous studies conducted in Los Angeles (Hudda et al.,
2010;Singh et al., 2006). Regarding volume concentrations, we observed one minor volume
mode at the size range of 300-500 nm and a major mode at around 4-6 µm. In this case, the
volume concentration was higher in the cold phase than in the warm phase for the minor
mode diameter (at 300-500 nm), while a sharper peak was observed for the major mode
diameter (at around 4-6 µm) in the warm phase compared to the cold phase. This PM volume
size distribution is typical of urban areas (Vu et al., 2015), and is also consistent with the
findings of a previous study conducted recently by this research group at the same sampling
location (Hasheminassab et al., 2014b).
*3.2. Number of Factors*
In the present work, the PMF model was run several times using different number of
factors, input uncertainty matrices (as noted in the methods section), and extra modeling
uncertainties to obtain the best and most physically applicable solution. Additionally, we used
several criteria to determine the best solution resolved by the model, including: 1) particle
number size distribution profiles for different factors; 2) volume size distribution profiles for
the resolved factors; 3) profiles of auxiliary variables for different factors; 4) contribution of
each factor in different seasons to the total number concentrations; 5) diurnal variations of





each of the resolved factors in the cold and warm phases; 6) diurnal variations of each of the
resolved factors in weekdays versus weekends; and 7) correlation between auxiliary variables
and the relative contribution of each of the resolved factors. The six-factor solution was
found to present the most physically explainable one, and was, therefore, chosen as the final
solution.  When the model was run with one less factor (i.e., 5-factor solution), the model
could not distinguish the two traffic factors, and Traffic 1 and Traffic 2 factors were merged
together. On the other hand, when the model was run with one more factor (i.e., 7-factor
solution), a new factor was resolved by the model , having  a mode diameter between that of
"urban background aerosol" and "secondary aerosol", but without having any distinct diurnal,
seasonal, or weekday/weekend trends or auxiliary variables profile. Therefore, this factor
could not be meaningfully interpreted and identified, prompting us to choose the 6-factor as
the optimal solution.
Figure 3 illustrates the number size distributions as well as the auxiliary variables
profiles for each of the factors resolved by the PMF. Figure 4 indicates volume size
distribution of each factor. In Figures 3, 4, and S1, the black solid lines represent absolute
concentrations (number or volume) of each size bin and should be read from the left Y axis,
while the grey triangles represent the explained variation of each size bin and should be read
from the right Y axis. The relative contributions (overall, and by cold or warm phases) of
each factor to the total number concentrations are shown in Figure 5. Figure 6 illustrates the
contribution (particles/cm$^3$) of each of the PMF-resolved factors to the total number
concentrations in the cold and warm phases within box and whisker plot. The diurnal
variations and the weekday/weekend trends (geometric means) for each of the factors are
illustrated in Figures 7 and 8, respectively. The spearman correlation coefficient matrix
indicating the association between the auxiliary variables and the factors resolved by the
PMF model is also presented in Table 3.
*3.3. Factor identification*
*Factor 1:* Factor 1 has a number mode at <20 nm, a volume mode at <20 nm, and
contributes 17.3% (11.7-24%)  to the total number concentrations (Figures 3, 4, and 5). This
factor has strong positive (except for RH, with which this factor has negative correlation)
associations with temperature, wind speed, SOC, and O$_3$ (Table 3), which are also
statistically significant ($p<0.05$). These associations are also apparent from high loadings of
temperature, RH, wind speed, and O$_3$ in the auxiliary variables profile (Figures 3 and S1).
The contribution of this factor to the total number concentration was also higher in the warm





phase than in the cold phase, when higher temperatures, wind speeds, and solar radiation are
observed (Figure 1); this was the case both in terms of percent contribution (24% in the warm
phase vs. 11.7% in the cold phase) and number concentration (589±25 particles/cm$^3$ in the
cold phase vs. 1153±28 particles/cm$^3$ in the warm phase) (Figures 5 and 6 and Table S1). The
diurnal variations for this factor also revealed a sharp peak in the afternoon (2-6 PM) (Figure
7), which coincides with very high temperatures, wind speeds, and solar radiation as well as
with minimum RH (Figure 1). However, there was no significant distinction in the diurnal
variation patterns of this factor in weekdays compared to weekends (Figure 8).

9       The above characteristics are all typical of a "nucleation" factor, during which new

particles are formed via photochemical events under high temperatures, high wind speeds,
and low RH (Beddows et al., 2015;Brines et al., 2015;Dall'Osto et al., 2012;Vu et al., 2015).
Our findings are most specifically consistent with those of the study of Brines et al. (2015), in
which the authors had reported nucleation as one of the major sources of UFPs in five high-
insolation cities, including Los Angeles using the data obtained from the same sampling
location. They observed very similar diurnal variation for nucleation, with peaks in early
afternoon at the same sampling location in Los Angeles.
*Factor 2:* Factor 2 is mostly represented by particles at 20-40 nm and contributes about
40% (33.2-43.4%) to the total number concentration (Figures 3 and 5). It also has a volume
concentration peak at around 30-40 nm (Figure 4). Judging by the loadings presented in the
auxiliary variables profile (Figures 3) and correlation coefficients presented in Table 3, this
factor has clear associations with gaseous pollutants (e.g., CO, NO, and NO$_2$), BC, EC, and
POC (from the scenario containing EC/OC data (Figure S1), which themselves are indicators
of vehicular emissions (Gu et al., 2011;Ogulei et al., 2006b). In addition, high species
loadings (Figure 3) and correlation coefficients (Table 3) of LDV and HDV counts can also
be observed for this factor, indicating the influence of nearby passing traffic on this factor.
The contribution of this factor to the total number concentration was also much higher in the
cold phase than in the warm phase, when lower temperatures, wind speeds, and solar
radiation (Figure 1) lead to increased atmospheric stability and lower mixing height
(Hasheminassab et al., 2014a); this was the case both in terms of percent contribution (43.4%
in the cold phase vs. 33.2% in the warm phase) and number concentration (3166±66
particles/cm$^3$ in the cold phase vs. 1201±61 particles/cm$^3$ in the warm phase) (Figures 5 and 6
and Table S1). The diurnal variations also revealed a distinctive pattern peaking in the
morning rush hours (around 7-8 AM) (Fig. 7). The weekday/weekend analysis also indicated





that this factor had higher contributions during the weekdays compared to the weekends
(Figure 8). Therefore, this factor can be attributed to traffic tailpipe emissions. Previous
source apportionment studies on number size distributions have also associated such
characteristics with fresh vehicular emissions (Beddows et al., 2015;Dall'Osto et al., 2012;Vu
et al., 2015). This factor is denoted as "traffic 1", given that another factor attributed to traffic
emissions was resolved, which will be discussed in the following section. The characteristics
of this traffic factor are in agreement with what Brines et al. (2015) reported for five high-
insolation cities, including Los Angeles.
*Factor 3:* This factor has a major peak in the Aitken mode (60-100 nm) and contributes
27.5% (25-27.6%) to the total number concentration (Figures 3 and 5). It also exhibited a
volume concentration peak at around 100 nm (Figure 4). Judging by the loadings presented in
the auxiliary variables profile (Figures 3 and S1) and correlation coefficients presented in
Table 3, significant associations can be observed between this factor and gaseous pollutants
(e.g., CO, NO, and $NO_2$), as well as with BC (and EC from the scenario containing EC/OC
data (Fig. S1)). Although weaker than those of Factor 2, there are significant positive
associations between this factor and LDV and HDV counts (Figure 3 and Table 3),
suggesting the likely influence of nearby passing traffic. This factor also had a significantly
higher contribution to the total number concentrations in the cold phase than in the warm
phase (an average of 1755±56 particles/$cm^3$ in the cold phase vs. 1059±43 particles/$cm^3$ in
the warm phase (Figure 6)), in spite of the fact that its percent contribution to the total PM
number concentrations was comparable in both phases, and slightly higher in the warm phase
(25% in the cold phase vs. 27.6% in the warm phase (Figure 5)). This is due mainly to the
fact that the contribution of the "traffic 1" factor is so large in the cold phase that has
significantly obscured the percent contribution of other factors in this phase, even though
their absolute contributions in terms of total number concentrations were higher in the cold
phase.
The diurnal variations for this factor also indicated clear peaks during the morning rush
hours (6-8 am) in both phases, along with another peak at late night during the cold phase,
most likely due to the stagnant atmospheric conditions during this time of the year, which
traps the emissions in lower altitudes (Figure 7). The weekday/weekend analysis for this
factor revealed larger contributions in weekdays than in weekends, especially during daytime
hours. The slightly higher nighttime contribution of this factor in the weekends compared to
the weekdays can be attributed to the larger number of within-city travels being made on





holiday nights. These levels and trends are, overall, suggestive of emissions from vehicular
sources. However, the larger size range of this factor compared to factor 2, combined with the
involvement of EC and SOC (as observed from the scenario containing EC/OC data (Fig.
S1)) as well as BC, suggest that although this factor also originates from "traffic", the
particles are "older" (i.e., more aged) than those observed in factor 2 and are mostly in the
Aitken and Accumulation modes; therefore, it was labeled as "traffic 2". This finding is also
consistent with those of the previous studies (e.g., Brines et al. (2015)), in which the authors
detected distinct traffic factors (with a collective relative contribution of approximately 60%
in Los Angeles at the same sampling site) using a different source apportionment method,
named k-means cluster analysis. It should be noted that it is quite common in source
apportionment studies performed on size-segregated PM number concentrations to detect
more than one traffic factors, due primarily to the fact that particle sizes may change, as
particles undergo processes including agglomeration as well as evaporation or condensation
of semi-volatile species from- or onto their surface following their release in the atmosphere
(Harrison et al., 2016;Kim et al., 2004;Zhou et al., 2005).
It is also noteworthy that the traffic 2 factor has a slightly higher HDV loading than
traffic 1 factor (Figures 3 and S1). It also has a somewhat stronger positive correlation with
HDV (R=0.43) than with LDV (R=0.41), while the traffic 1 factor has a stronger correlation
with LDV (R=0.69) than with HDV (R=0.52). Additionally, the stronger correlation of traffic
2 factor with EC and BC compared to traffic 1 leads us to the hypothesis that HDV might be
contributing more to this factor than LDV is. Vu et al. (2015) have also suggested that
observing a number concentration mode at the size range of 60-100 nm can be a result of
incomplete combustion of diesel fuel, consisting of pyrolytic EC and OC. Other studies have
also found two particle modes, or factors, for traffic-related emissions. Although the
emissions in both of these two modes are believed to come from the same fleet of vehicles,
they have different formation mechanisms and chemistry, with particles associated with the
second mode (i.e., soot mode) assumed to have an elemental carbon core. This is consistent
with the findings of the present study, judging by the mode diameter and high loading of BC,
EC, and OC in the traffic 2 factor (Figures 3 and S1) and the strong correlation of this factor
with BC, EC, and OC (Table 3). Additionally, studies have indicated that a fraction of diesel
PM emissions, which is generally in the range of 50-200 nm, comprises particles that have an
elemental core, with low-vapor-pressure hydrocarbons and sulfur compounds being adsorbed
on their surface (Burtscher, 2005). Therefore, it might be likely that this factor is representing



a higher contribution of HDV emissions, although stronger evidence is required to confirm
this hypothesis.
*Factor 4:* Factor 4, which contributes 12.2% (7.4-17.1%) to the total number
concentration, is represented by a number mode at around 220 nm and a volume mode at
around 250 nm (Figures 3, 4, and 5). The profile for the auxiliary variables also indicates
high loadings for gaseous pollutants (e.g., CO, NO, and $NO_2$) and BC (Figure 3) as well as
for EC and SOC (when the PMF model was run with the EC/OC data (Fig. S1)). The large
correlation coefficients of this factor with the aforementioned species also confirm its strong
association with these parameters (Table 3). The lower-than-unity $NO/NO_2$ ratio for this
factor also suggests that these particles are aged compared to the newly formed particles (Liu
et al., 2014). This is also supported by the stronger positive correlation of this factor with
SOC than with POC, suggesting the fact that this factor is not coming from direct emissions
and has most likely undergone processes and reactions in the atmosphere. As can be inferred
from Figures 5 and 6, the contribution of this factor is significantly higher in the cold phase
than in the warm phase, both in terms of percent contribution (17.1% in the cold phase vs.
7.4% in the warm phase) and the absolute contribution to the total number concentration
($1200\pm41$ particles/cm$^3$ in the cold phase vs. $284\pm23$ particles/cm$^3$ in the warm phase). As
seen in Figure 7, the diurnal variations for this factor also exhibit a clear peak during morning
hours, which indicates higher concentrations when atmosphere is more stable and wind
speeds are low, especially in the cold phase when these conditions are even more intense
(Figure 1). The weekday/weekend analysis also revealed a slightly elevated contribution of
this factor to the total number concentrations during morning rush hours, especially during
the weekdays, suggesting the small influence of traffic emissions on this factor. Previous
studies have indicated that these are characteristics of the "urban background aerosol", as
suggested by (Beddows et al., 2015;Dall'Osto et al., 2012).
*Factor 5:* Factor 5 has a number and volume mode at around 500 nm and a minor
number mode at 50 nm (looking at the black dots, representing the explained variations)
(Figures 3 and 4). This factor contributes 2.1% (1.5-2.5%) to the total number concentration
(Figure 5). It is also associated with high loadings of $PM_{2.5}$ mass concentration (i.e., major
contributor to $PM_{2.5}$ mass), NO, $NO_2$, Temperature, RH (Figure 3), and SOC (as observed
from the scenario containing EC/OC data (Figure S1)). This is also supported by the results
of the correlation analysis presented in Table 3, indicating that this factor has strong positive




correlations with $PM_{2.5}$, NO, $NO_2$, Temperature, RH, and SOC. The overall small
contribution of this factor to the total number concentration was slightly higher in the cold
phase than in the warm phase; this was the case both in terms of percent contribution (2.5%
in the cold phase vs. 1.5% in the warm phase) and number concentration (111±11
particles/$cm^3$ in the cold phase vs. 100±5 particles/$cm^3$ in the warm phase) (Figures 5 and 6
and Table S1). The diurnal variation for this factor also reveals a significant increase during
nighttime, especially during the cold phase (Figure 7). However, the weekday/weekend
analysis did not reveal any distinctive trend pertaining to the day of the week for this factor
(Figure 8). These pieces of evidence point to "secondary aerosols" as the most appropriate
title for this factor, which is consistent with the results of previous PMF studies both on
number size distributions and chemical speciation data (Beddows et al., 2015;Hasheminassab
et al., 2014a). Table 3 indicates a  much higher correlation of this factor with SOC than POC
(R values of 0.5 and 0.2, respectively). The association of this factor with RH and
temperature, along with its higher contribution to particle number during the cold phase,
particularly at night, support the hypothesis that this factor likely represents the fraction of
aerosols produced by secondary reactions on a regional scale, including ammonium nitrate
(whose partitioning in the PM phase increases with decreasing temperature and increased
RH), but also secondary organic aerosols from nighttime and/or aqueous phase reactions, as
indicated in earlier studies in this area (Hersey et al., 2011;Venkatachari et al., 2005). In a
previous source apportionment study on $PM_{2.5}$ chemical speciation data in downtown Los
Angeles, we also found a similar factor profile, representing a mixture of secondary
components (dominated by secondary nitrate and SOC) with higher contribution during the
cold season (Hasheminassab et al., 2014a). Moreover, previous studies have shown that
secondary organic aerosol formed at nighttime together with ammonium nitrate are major
contributors to the mass concentrations of $PM_{2.5}$, which was also observed in the present
work from the high loading of $PM_{2.5}$ mass concentration in this profile (Figure 3)
(Hasheminassab et al., 2014a;Arhami et al., 2010;Saffari et al., 2016).
*Factor 6:* Factor 6 is dominated by particles at around 1 μm and above (Figure 3). This
factor also had a volume mode at > 1 μm (Figure 4). Although this factor contributes only
1.1% (0.2-6.3%) to the total number concentration (Figure 5), it is associated with high
loadings of coarse PM and $PM_{2.5}$ (great contributor to mass) (Figure 3). In addition, high
loadings of temperature and wind speed were observed for this factor (Figure 3). Table 3 also
indicates strong correlation of this factor with coarse PM, $PM_{2.5}$, temperature, and wind





speed. The contribution of this factor to the total number concentration was also higher in the
warm phase than in the cold phase, both in terms of percent contribution (0.2% in the cold
phase vs. 6.3% in the warm phase) and number concentration ($14\pm1$ particles/cm$^3$ in the cold
phase vs. $243\pm3$ particles/cm$^3$ in the warm phase) (Figures 5 and 6 and Table S1). The diurnal
variations for this factor exhibited significantly higher contributions during daytime,
especially in the warm phase (Figure 7), when atmosphere is unstable, wind speed is high,
and the mixing height is at its maximum (Figure 1). However, the weekday/weekend analysis
did not reveal any distinctive trend pertaining to the day of the week for this factor (Figure 8).
Based on all of the abovementioned characteristics, this factor was named "soil/road dust"
(Gietl et al., 2010;Harrison and Booker, 2001;Harrison et al., 2012). This is also quite
consistent with the findings of the study of Hasheminassab et al. (2014a), in which the
authors apportioned the sources of ambient fine particulate matter across the state of
California. In that study, the authors observed a lower contribution of the soil factor to
particle mass concentrations in the northern regions of the state of California, mainly because
of higher RH and increased precipitation that inhibit the re-suspension of soil due to strong
winds (Harrison and Booker, 2001). In the present study, similarly, the contribution of this
factor was higher in the warm phase, when higher temperatures and wind speeds facilitate the
re-suspension of soil and dust (Figure 1).
**4. Summary and conclusions**

21        The present study was the first attempt to characterize major sources of PM number

concentrations and quantify their contributions using the PMF receptor model applied on PM
number size distributions in the range of 13 nm to 10 μm combined with several auxiliary
variables, including BC, EC/OC, PM mass, gaseous pollutants, meteorological, and traffic
flow data, in central Los Angeles. The six-factor solution was found to be the most physically
applicable solution for the input data: nucleation, traffic 1, traffic 2, urban background
aerosol, secondary aerosol, and soil. Traffic sources (1 and 2) were the major contributor to
PM number concentrations, making up to above 60% of the total number concentrations
combined, with larger contributions in the cold phase compared to the warm phase, when
lower temperatures, wind speeds, and solar radiation lead to increased atmospheric stability
and lower mixing height. The contribution of traffic factors was largest during morning and
afternoon rush hours; it was also higher in the weekdays compared to the weekends, as
expected. In agreement with the findings of previous studies in Los Angeles, nucleation was





another major factor contributing to the total number concentrations (17%), having a larger
contribution in the warm phase than in the cold phase. The diurnal variations for this factor
also revealed a sharp peak in the afternoon (2-6 PM), which coincides with high
temperatures, wind speeds, and solar radiation as well as with minimum RH, providing ideal
conditions for the occurrence of photochemical nucleation processes, especially during
warmer seasons. Urban background aerosol, secondary aerosol, and soil, with relative
contributions of approximately 12%, 2.1%, and 1.1%, respectively, overall accounted for
approximately 15% of PM number concentrations. However, these factors dominated the PM
volume and mass concentrations, due mainly to their larger mode diameters.
**Acknowledgement**
The authors wish to acknowledge the support from the USC Viterbi School of Engineering's
Ph.D. fellowship award.

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



Table 1. Summary of the input parameters to the PMF model in this study.

| Parameter | Source of data | Time resolution in original data set |
|---|---|---|
| EC, OC | Sunset monitor | 1 hr |
| Size Distribution (14-760 nm) | SMPS | 5 min |
| Size Distribution (0.3-10 μm) | OPS | 5 min |
| BC | Aethalometer | 15 min |
| PM mass concentration data ($PM_{10-2.5}$, $PM_{2.5}$) | CARB | 1 hr |
| Gaseous Pollutants (NO, $NO_2$, CO, $O_3$, $SO_2$) | CARB | 1 hr |
| Meteorological data (T, RH, WS) | CARB | 1 hr |
| Traffic data (counts of LDV and HDV) | PeMS | 1 hr |





Table 2. Summary statistics for the parameters included in the PMF model.

| Species | Geometric Mean | Standard Error | Min | Max | S/N ratio |
|---|---|---|---|---|---|
| Total number concentration (#/cm$^{-3}$) | 6860.00 | 94.10 | 524.00 | 32400.00 | 7.00 |
| PM$_{10-2.5}$ (µg/m$^3$) | 15.90 | 0.19 | 2.00 | 77.00 | 7.00 |
| PM$_{2.5}$ (µg/m$^3$) | 14.50 | 0.23 | 1.00 | 101.00 | 6.90 |
| CO (ppm) | 0.58 | 0.01 | 0.10 | 2.19 | 7.10 |
| NO (ppb) | 8.46 | 0.57 | 1.00 | 212.00 | 5.80 |
| NO$_2$ (ppb) | 22.50 | 0.23 | 1.90 | 75.00 | 7.10 |
| O$_3$ (ppb) | 17.40 | 0.33 | 2.00 | 105.00 | 6.80 |
| BC (µg/m$^3$) | 1.14 | 0.02 | 0.124 | 9.13 | 6.90 |
| POC*(µg/m$^3$) | 2.20 | 0.08 | 0.10 | 19.20 | 6.80 |
| SOC* (µg/m$^3$) | 2.13 | 0.05 | 0.04 | 16.30 | 7.10 |
| EC* (µg/m$^3$) | 1.01 | 0.03 | 0.01 | 7.34 | 8.80 |
| RH (%) | 50.40 | 0.40 | 6.00 | 99.00 | 7.10 |
| Temperature (°C) | 18.80 | 0.13 | 3.89 | 38.33 | 7.30 |
| Wind speed (m/s) | 4.03 | 0.04 | 1.00 | 14.00 | 6.80 |
| LDV (#/h) | 3790 | 34 | 691 | 7620 | 7.10 |
| HDV (#/h) | 153 | 3 | 5 | 920 | 6.80 |

*Values are pertaining to the runs including EC/OC data.



Table 3. Spearman correlation coefficient matrix indicating the association between the
auxiliary variables and the factors resolved by the PMF model. R values above 0.5 are
bolded.

| Species | Nucleation | Traffic 1 | Traffic 2 | Urban Background Aerosol | Secondary Aerosol | Soil/Road Dust |
|---|---|---|---|---|---|---|
| $PM_{10-2.5}$ | 0.17* | 0.21* | 0.35* | 0.27* | 0.09* | 0.39* |
| $PM_{2.5}$ | -0.24* | -0.09* | 0.05* | 0.33* | **0.69*** | 0.23* |
| CO | 0.04 | 0.41* | **0.58*** | **0.64*** | 0.17* | 0.28* |
| NO | -0.01 | 0.48* | **0.59*** | **0.52*** | 0.27* | 0.24* |
| $NO_2$ | 0.08* | **0.50*** | **0.60*** | **0.57*** | 0.33* | 0.14* |
| $O_3$ | **0.57*** | 0.34* | 0.40* | -0.35* | 0.46* | 0.19* |
| BC | 0.01 | **0.53*** | **0.70*** | **0.71*** | 0.13* | 0.22* |
| POC | 0.09* | **0.62*** | 0.28* | 0.30* | 0.24* | 0.29* |
| SOC | 0.46* | 0.12* | 0.43* | **0.58*** | 0.46* | 0.20* |
| EC | 0.17* | 0.47* | **0.56*** | **0.60*** | 0.20* | 0.17* |
| RH | -0.26* | -0.32* | -0.30* | -0.05* | 0.43* | 0.33* |
| Temp | **0.52*** | -0.23* | -0.18* | -0.39* | 0.34* | 0.47* |
| WS | **0.57*** | 0.00 | 0.07* | -0.04* | -0.25* | **0.62*** |
| LDV | 0.22* | **0.70*** | 0.42* | 0.05* | 0.01 | 0.02 |
| HDV | 0.23* | **0.52** | 0.43* | -0.08* | -0.12* | -0.01 |

* Indicates R values that are statistically significant (P<0.05).





1    Figure 1. Diurnal variations of important meteorological parameters in the cold and warm
2    phases. Error bars correspond to one standard error.

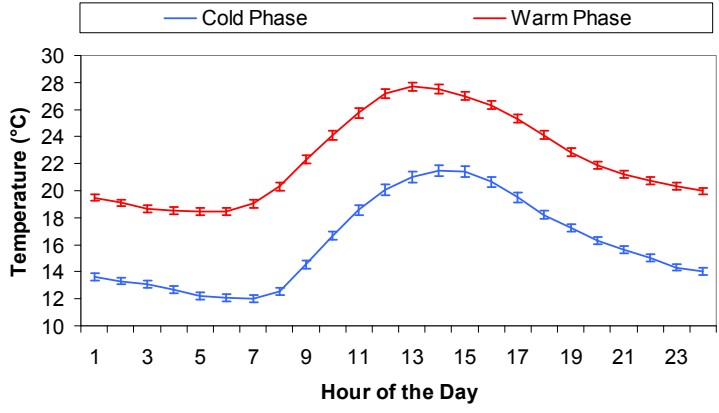

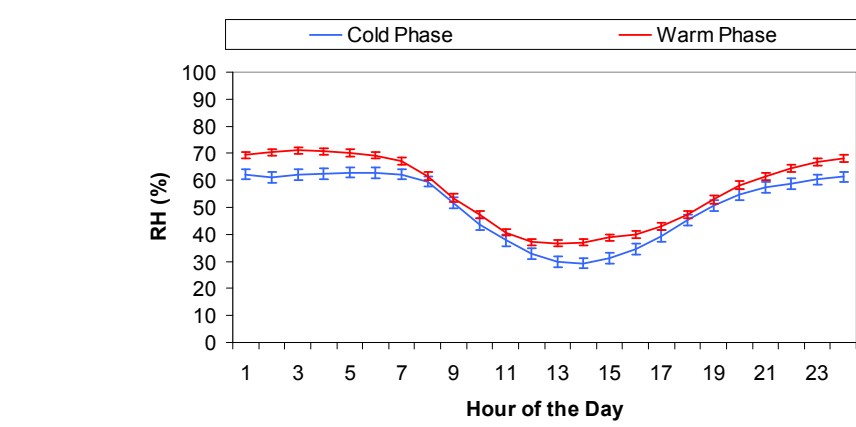

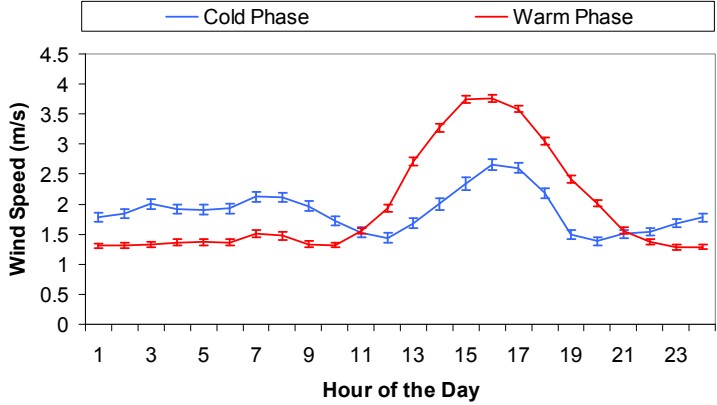





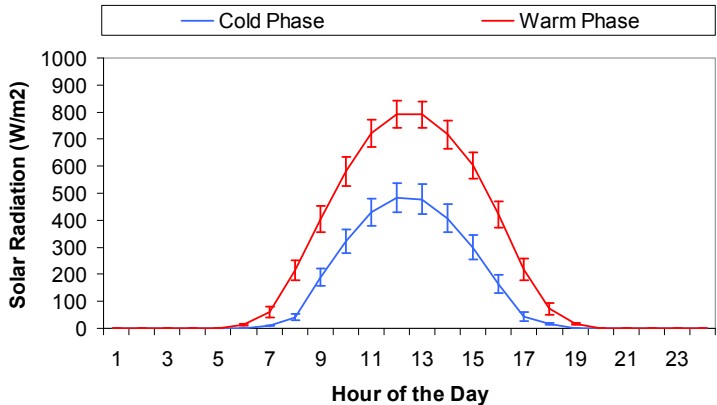






Figure 2. Average number and volume size distributions of all the input samples to the PMF
model in the cold and warm phases (the graphs represent geometric means ± SE).

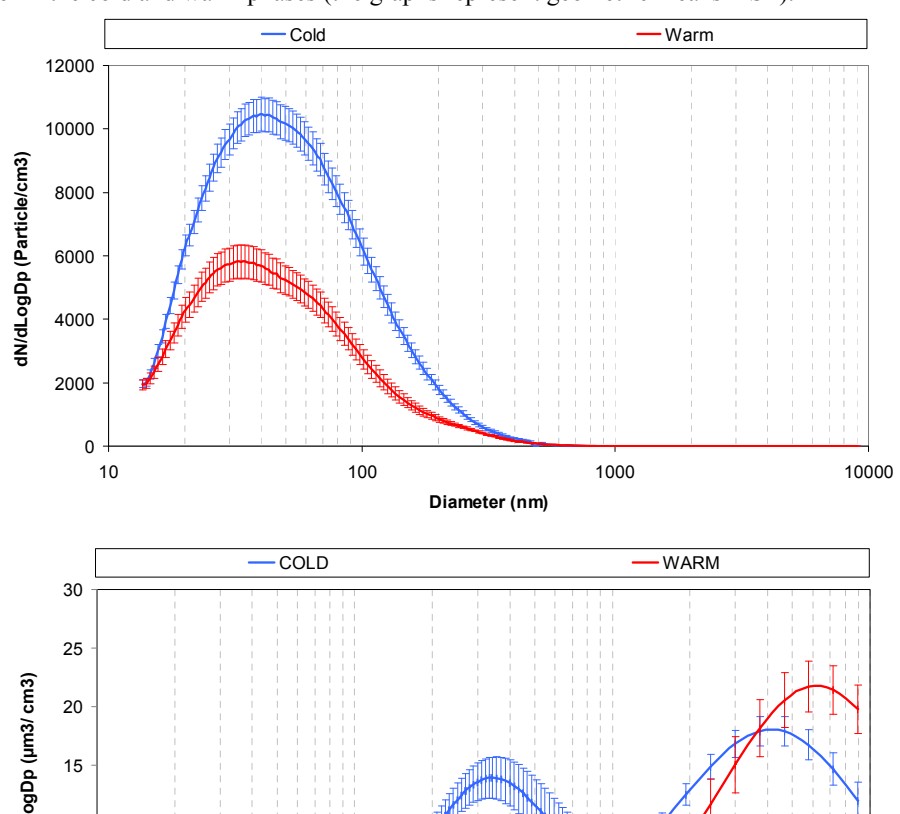

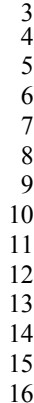





Figure 3. The number size distributions as well as the auxiliary variables profiles for each of the factors resolved by the PMF model.

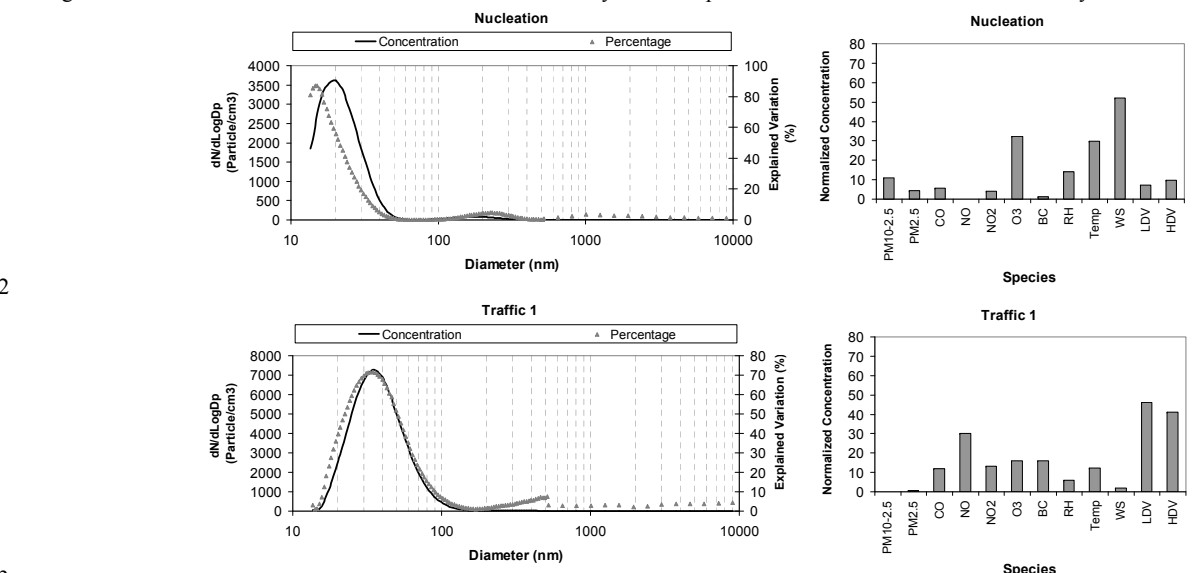





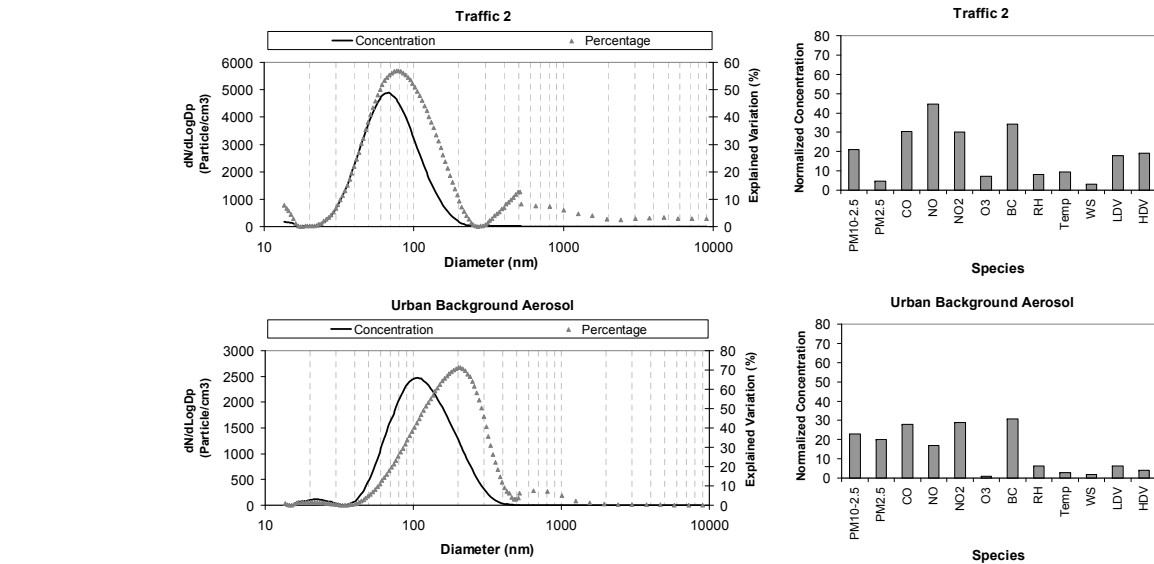

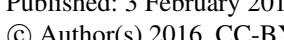
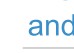
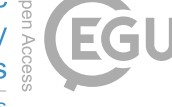


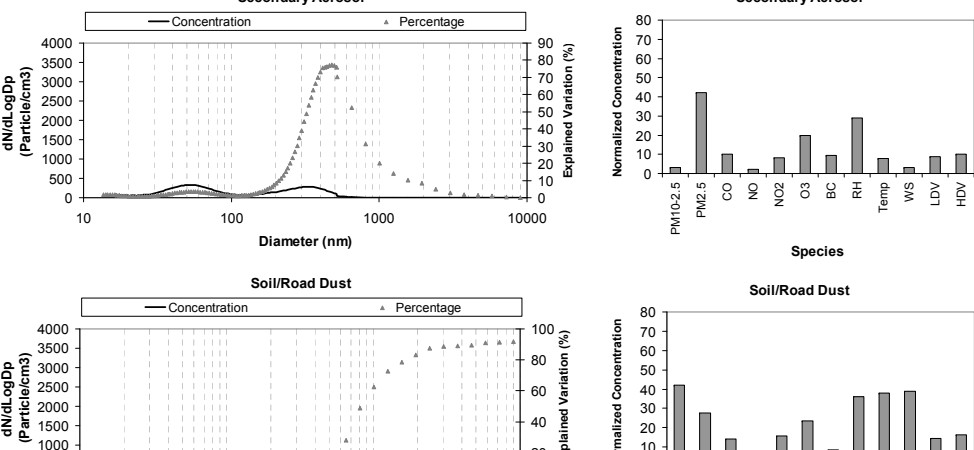






Figure 4. Volume size distributions along with the explained variation (%) of each factor
profile resolved by the PMF model.









Figure 5. Relative contribution of each factor to the total number concentrations: a) overall
phases; b) cold phase; and c) warm phase.

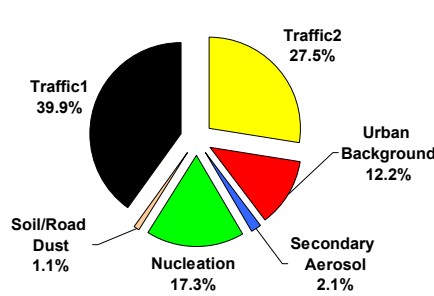

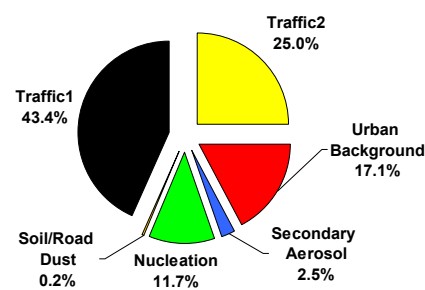

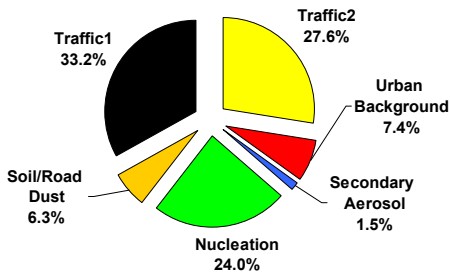






Figure 6. Contribution (particles/cm$^3$) of each of the PMF-resolved factors to the total number
concentrations in the cold and warm phases.

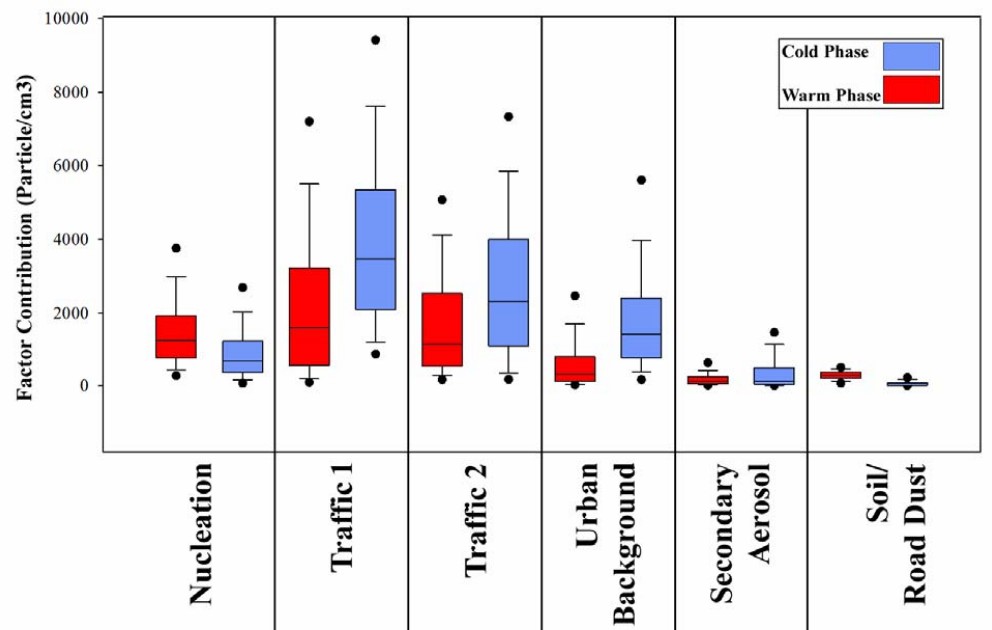





Figure 7. Diurnal variations (geometric means) of number concentrations (particles /cm³)
from each factor resolved by the PMF model in the cold and warm phases.  Error bars
correspond to one standard error.

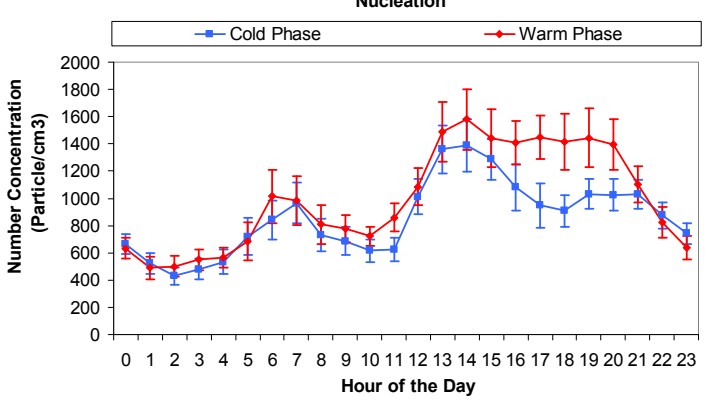

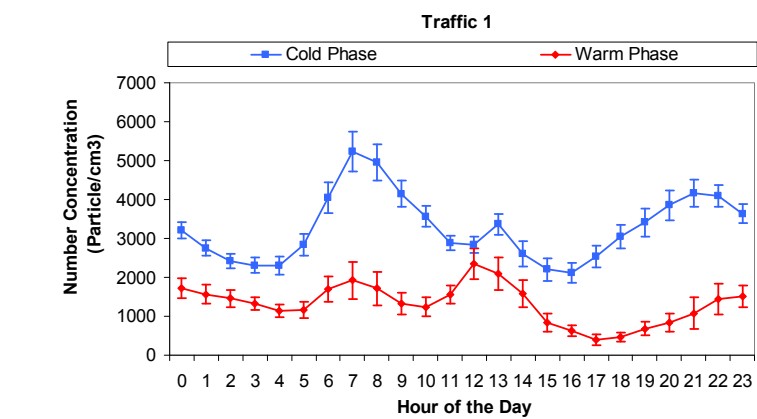

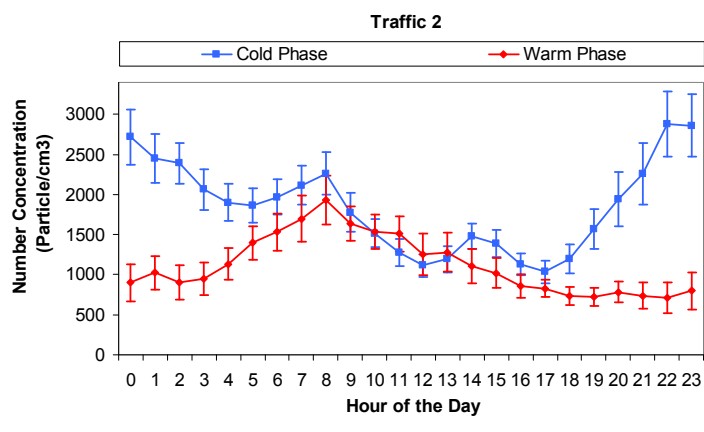



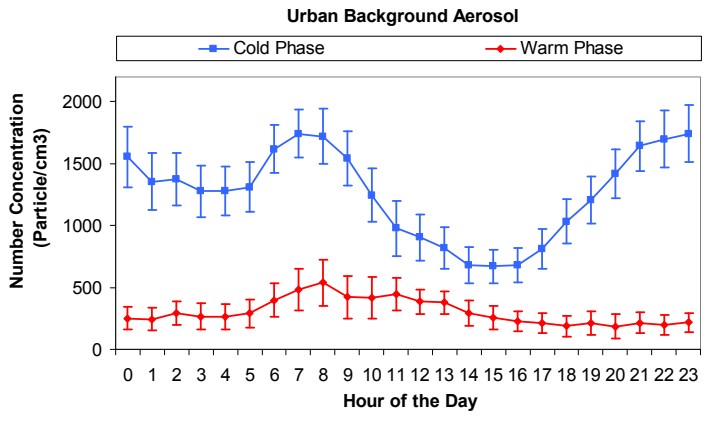

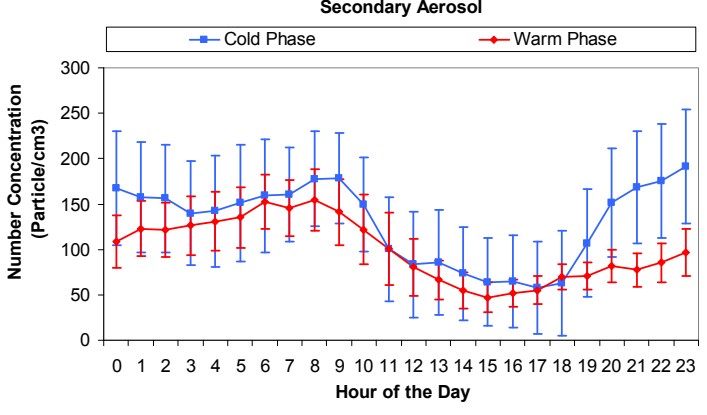

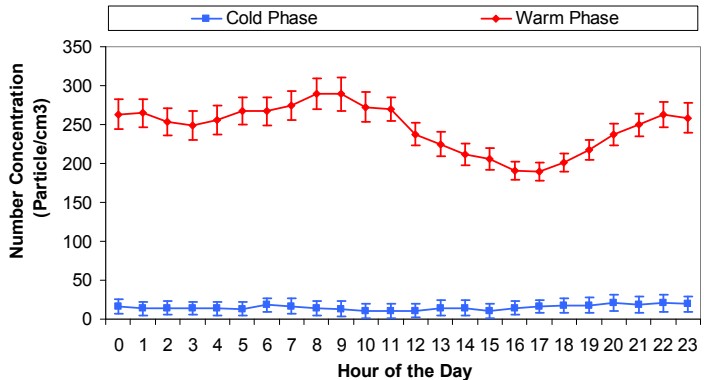





Figure 8. Weekday/weekend analysis of each of the factors resolved by the PMF model
(values are geometric means). Error bars correspond to one standard error.

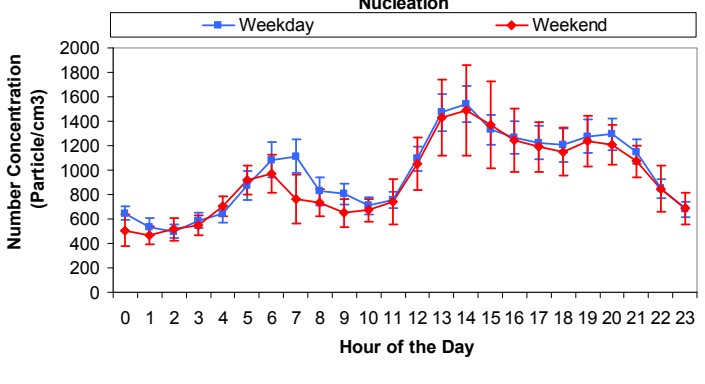

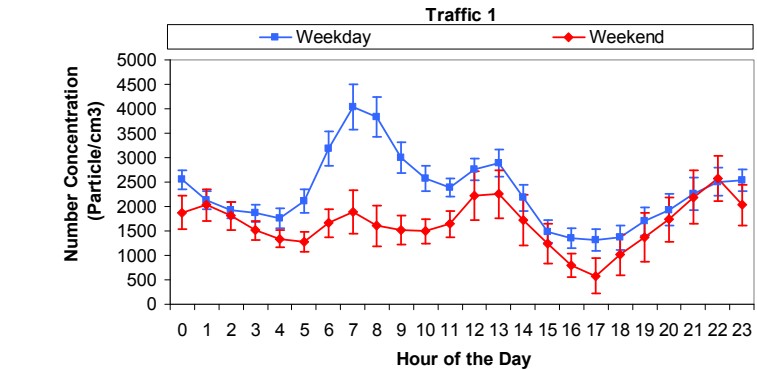

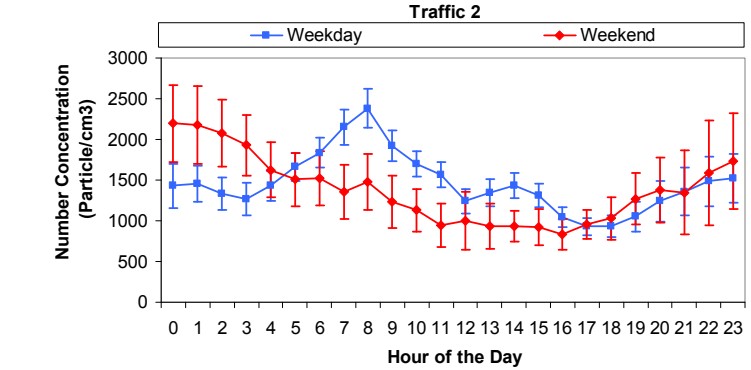



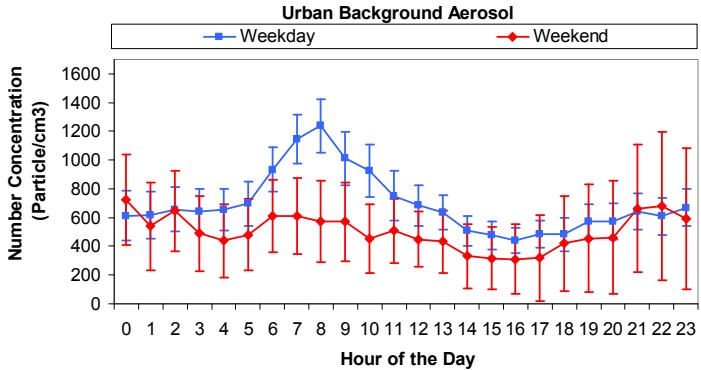

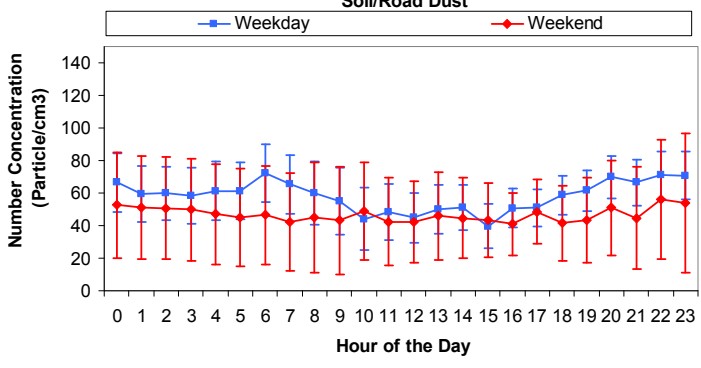

