# Peer review of "Source apportionment of ambient particle number concentrations in central Los Angeles using positive matrix factorization (PMF)"

_Atmospheric Chemistry and Physics, 2015_

## Referee Comment (RC1) · Anonymous Referee #1 · 1 Mar 2016

SOURCE APPORTIONMENT OF PARTICLE NUMBERS IN LOS ANGELES There is very widespread interest currently in particle number concentrations in the atmosphere, and the determinants of those concentrations. Much of the recent research has focused on the global atmosphere, with particular attention being paid to nucleation processes. While such research is important in the context of global climate, there is also a very important research area associated with particulate matter in the urban atmosphere. While a huge number of source apportionment studies of particle mass have been carried out in urban environments around the world, there has been a relative scarcity of source apportionment of particle number concentrations. The latter requires data sets which are not available from many locations, and the rather challeng-

ing application of source apportionment programs. Consequently, there are few high quality studies in the literature and this paper presents a valuable addition for a major city with a well characterised air pollution climate. It can be very challenging to extract meaningful factors from PMF, but the authors appear to have succeeded in doing so, and have provided convincing associations with the sources to which they attribute the factors. In summary, the data interpretation appears to be sound, and consequently the paper offers excellent quantitative insights into the sources contributing to the particle number count as a function of particle size within Los Angeles. There are some relatively minor points which need to be addressed, relating to both the methodology and the data interpretation. These are as follows: A. At no point is it clearly stated how the particle size distribution data are sub-divided for uploading to the PMF. Were the size bins from the SMPS used without subsequent combination into larger size fractions and what size ranges were used from the OPC? There is also a question as to how the errors in the SMPS and OPC data were quantified in order to populate the error matrix of the PMF. These are points of interest which should be included in the paper.

B. The PMF does not appear to have separated the factors wholly cleanly (which is not unusual). Thus, the nucleation factor (Factor 1) appears to show some influence of morning rush hour traffic seen in Figures 7 and 8. A comment on the magnitude of this overlap would be useful. The nucleation factor has significant magnitude, even in the depths of the night, which require some explanation. Is this a reflection that nucleation continues through the night, or is it an artefact of the PMF?

C. The Traffic 2 factor shows a rather odd diurnal variation, which differs substantially between the cold phase and the warm phase and between weekday and weekend data. The diurnal profiles seem to suggest a substantial contribution of semi-volatile materials, especially during the cold phase, as reflected in the substantial increase at night time.

D. The huge difference between the cold phase and the warm phase seen in Figure 7

for the soil/road dust factor is rather surprising. Are the weather conditions really that different between the cold phase and the warm phase, such that they can explain such a huge seasonality?

E. Some recent papers have attributed a significant proportion of particulate matter mass in California to emissions from cooking. Such attribution usually comes from AMS data, largely on the basis of diurnal profiles. Is there any indication of such a source within this data set? One might expect it to be more prominent at the weekend than on weekdays, especially if related to outdoor barbequing.

Minor points which require attention are as follows: F. Page 3, line 6. This implies that regulations on PM number emissions from road vehicles in Europe have been set on the basis of health studies of UFPs. This is not the case. The PMP number limits, which have been introduced as part of recent European vehicle emission standards, arose because of the difficulty encountered in achieving repeatable measurements of very low mass concentrations of PM in diluted engine exhaust, whereas particle number could be measured very repeatedly after removable of the semi-volatile fraction. Hence a test procedure, based upon solid particle number, was considered a more practicable option than determination of mass.

G. Page 5, line 34. Last word should read 'factor' rather than 'actor'.

H. Page 6, line 30. 'Ration' should read 'ratio'.

I. It would be good to quote the OC/EC gradient determined through equation 1, as there is still considerable interest in applications of the EC Tracer Method to source apportionment of organic carbon.
* * *

---

## Referee Comment (RC2) · Anonymous Referee #2 · 2 Mar 2016

The study is focused on the characterization of the major sources of PM number concentrations and on the quantification of their contributions using the PMF receptor model applied to PM number size distributions combined with several auxiliary variables in central Los Angeles. The topic is interesting, the data set is large and reliable, the paper is well organized and data interpretation seems to be sound. While there are many articles regarding aerosol source identification by PMF, there are few regarding the analysis of particle number concentrations by PMF. This work gives a very good quantitative identification of the sources which contribute to the particle number as a function of particle size in Los Angeles. I have only few minor remarks: 1) p 14 l.32-34: why the same explanation is not valid for traffic 1?  2) Factor 5: is there any explanation why this factor gives such a small contribution to particle number? This factor is attributed to secondary nitrates and organics (quite reasonable). Is not there any contribution from secondary sulfates in Los Angeles? If yes, in which of the identified factors is? 3) p 18 l.9: the factor was called "soil/road dust" but the time trend of this source does not justify road dust as a source 4) Fig 3: the normalized concentrations of PM2.5-10 are higher respect to PM2.5 for both traffic 2 and urban background; is there any possible explanation? 5) Fig. 8: why on weekend there is a night peak only for traffic 2?

---

## Author Comment (AC1) · 11 Mar 2016

**Reviewer #1:**

**SOURCE APPORTIONMENT OF PARTICLE NUMBERS IN LOS ANGELES**
**There is very widespread interest currently in particle number concentrations in the atmosphere, and the determinants of those concentrations. Much of the recent research has focused on the global atmosphere, with particular attention being paid to nucleation processes. While such research is important in the context of global climate, there is also a very important research area associated with particulate matter in the urban atmosphere. While a huge number of source apportionment studies of particle mass have been carried out in urban environments around the world, there has been a relative scarcity of source apportionment of particle number concentrations. The latter requires data sets which are not available from many locations, and the rather challenging application of source apportionment programs. Consequently, there are few high quality studies in the literature and this paper presents a valuable addition for a major city with a well characterized air pollution climate. It can be very challenging to extract meaningful factors from PMF, but the authors appear to have succeeded in doing so, and have provided convincing associations with the sources to which they attribute the factors. In summary, the data interpretation appears to be sound, and consequently the paper offers excellent quantitative insights into the sources contributing to the particle number count as a function of particle size within Los Angeles. There are some relatively minor points which need to be addressed, relating to both the methodology and the data interpretation.**

**Authors**: We thank the reviewer for his/her valuable comments on the paper that has improved the quality of the work. Please find below our detailed response and the modifications made to the manuscript according to each comment.

**(A) At no point is it clearly stated how the particle size distribution data are subdivided for uploading to the PMF. Were the size bins from the SMPS used without subsequent combination into larger size fractions and what size ranges were used from the OPC? There is also a question as to how the errors in the SMPS and OPC**

**data were quantified in order to populate the error matrix of the PMF. These are points of interest which should be included in the paper.**

**Response**: The reviewer's comment is properly taken and carefully addressed as discussed in detail below.

Regarding the first part of the reviewer's question (i.e. the size bins used from SMPS and OPS), it should be mentioned that the size bins from the SMPS were used without subsequent combination into larger size fractions. Additionally, the SMPS size bins from 13.6-514 nm were merged with the OPS size channels covering 0.522-9.01 μm as the input data to the PMF model. This information is now added in the revised manuscript (Page 6, Lines 3-5):

"Therefore, size bins covering the range of 13.6-514 nm from SMPS (without subsequent combination into larger size fractions) were merged with the OPS channels from 0.522 to 9.01 μm as the input data to the PMF model. "

Regarding the second part of the comment (i.e. the estimation of errors from the SMPS and OPS), we had already provided detailed information as to how these errors were estimated and used to calculate the uncertainties associated with each single data point (Page 9, Lines 7-24):

"In the present work, since no measurement uncertainties were available for the input parameters, we applied the method suggested by Ogulei et al. (2006a;2006b) and Zhou et al. (2014) to calculate the uncertainties for individual data points inserted into the model. For this purpose, measurement errors were first estimated for each data point using the following equation:

$$\sigma_{ij} = C_1 (N_{ij} + \overline{N}_j) \qquad (6)$$

where, $\sigma_{ij}$ is the estimated measurement error for the $i$th sample and $j$th size bin (or concentration of auxiliary variables); $C_1$ is an empirical constant usually between 0.01 and 0.05; $N_{ij}$ is the observed number concentration for the $i$th sample and $j$th size bin (or concentration of auxiliary variables); and $\overline{N}_j$ is the arithmetic mean of the PM number concentrations for the $j$th size bin (or concentration of auxiliary variables).

The value of the measurement method obtained from the above equation is then used to calculate the measurement uncertainty, according to the following equation:

$$S_{ij} = \sigma_{ij} + C_2 \max(|x_{ij}|, |y_{ij}|) \qquad (7)$$

where, $S_{ij}$ is the calculated uncertainty associated with the $i$th sample and $j$th size bin; $C_2$ is an empirical constant usually between 0.1 and 0.5; and $Y_{ij}$ is the value calculated by the model for $x_{ij}$. In the present work, $C_1$ and $C_2$ values of 0.05 and 0.1 were chosen to obtain the most physically interpretable solution using a trial and error approach."

It should be noted that this method has been successfully used in a large number of studies on PM source apportionment using particle number size distribution, including but not limited to (Beddows et al., 2015;Friend et al., 2012;Harrison et al., 2011;Krecl et al., 2008;Ogulei et al., 2007;Ogulei et al., 2006a;Ogulei et al., 2006b).

**(B1) The PMF does not appear to have separated the factors wholly cleanly (which is not unusual). Thus, the nucleation factor (Factor 1) appears to show some influence of morning rush hour traffic seen in Figures 7 and 8. A comment on the magnitude of this overlap would be useful.**

**(B2) The nucleation factor has significant magnitude, even in the depths of the night, which require some explanation. Is this a reflection that nucleation continues through the night, or is it an artefact of the PMF?**

**Response**: The reviewer's comment is properly taken and the response is provided in detail below.

**(B1)** We do not believe that the minor peak observed in the morning rush hours in the nucleation factor is due to PMF artifact. Although the maximum amount of nucleation is expected to occur in early afternoon, when temperature and solar radiation are highest leading to maximum photochemical activity in the atmosphere, we expect to observe some influence of nucleation during morning rush hours due to the cooling, following dilution, of vehicular emissions  and the partitioning of semi-volatile species into the

particle phase (Harrison et al., 2011). A Similar trend was observed by Harrison et al. (2011) in number size distributions on data collected near a major road way in central London. The "nucleation" factor resolved by Harrison et al. (2011) exhibited even a larger peak in the morning rush hours than that observed in the early afternoon, mainly because their measurements were carried out near a major roadway, while in the present study measurements were performed in an urban background location that is affected by vehicular emissions. Similar to what was observed in our study, the authors argued that this nucleation peak was due to the dilution of diesel exhaust emissions in the low temperatures observed during that time of day, and has been supported by the results from other studies, including (Charron and Harrison, 2003;Janhäll et al., 2004;Ntziachristos et al., 2007). In addition, our results are also consistent with those from the study of Brines et al. (2016), in which the authors observed very similar diurnal variation for nucleation, with a minor peak in the early morning and a major peak in early afternoon at the same sampling location in Los Angeles, using the k-means clustering approach for PM number apportionment.

Therefore, to fully address the reviewer's comment we have added the following text to the revised manuscript:

Page 13, Lines 12-14: "A minor peak was also observed during morning rush hours (6-8 am), which suggests the partial influence from traffic sources, as also observed by loadings of HDV and LDV in this factor (Figure 3)."

Page 13, Lines 20-23: " The minor peak in the early morning can also be explained by the cooling, following dilution, of vehicular exhaust emissions, which leads to the partitioning of semi-volatile exhaust gases into the particle phase; this process is further enhanced by to the lower temperatures during that time of day (Harrison et al. 2011; Charron and Harrison, 2003; Janhall et al. 2004; Ntziachristos et al. 2007)."

Page 13, Lines 26-28: "They observed very similar diurnal variation for nucleation, with a minor peak in the early morning and a major peak in early afternoon at the same sampling location in Los Angeles."

**(B2)** In regards to the second part of the comment, we believe that it may have been due to both the PMF artifacts and occurrence of low levels of nucleation during the night. As the respected reviewer has mentioned, it is not unusual for the PMF model to fail to

cleanly resolve different factors, and it is likely that the trends do not exactly follow the characteristics that are expected from the resolved factor; thus, they are assigned to PMF artifacts. This can also be the case here, as can be true for all other PMF source apportionment studies. On the other hand, looking at the results of previous studies, it is unlikely that the contribution of a factor such as nucleation would go down to zero after peaking at specific times of the day (i.e., early morning and early afternoon). For example, Harrison et al. (2011) presented a diurnal variation chart for the nucleation factor, peaking in early morning and early afternoon hours, but not decreasing down to very low levels following these peaks or during nighttime. As can be seen in Figure 7, the contributions of nucleation factor are reduced substantially to lower levels at night, reaching background levels, but are still detectable. Moreover, the diurnal trend of nucleation factor in this study is in very good agreement with the findings of Brines et al. (2016), in which the authors reported nucleation as one of the major sources of UFPs in Los Angeles using the data obtained from the same sampling location.

**(C) The Traffic 2 factor shows a rather odd diurnal variation, which differs substantially between the cold phase and the warm phase and between weekday and weekend data. The diurnal profiles seem to suggest a substantial contribution of semi-volatile materials, especially during the cold phase, as reflected in the substantial increase at nighttime.**

**Response**: The reviewer's comment is properly taken and the following sentence has been added to the manuscript text:
Page 15, Lines 10-12 "This diurnal profile suggests a major contribution from semi-volatile compounds in the atmosphere, particularly in the cold phase, as reflected in the substantial increase at nighttime."

In addition, more detailed discussion on the characteristics of Traffic 2 and its diurnal variability, particularly during weekday vs. weekend, have been presented in response to comment 1 of the second reviewer.

**(D) The huge difference between the cold phase and the warm phase seen in Figure 7 for the soil/road dust factor is rather surprising. Are the weather conditions really that different between the cold phase and the warm phase, such that they can explain such a huge seasonality?**

**Response:** The reviewer's comment is properly taken and the response is provided in detail below.

Based on Figure 1, it can be observed that the meteorological conditions are indeed different in the cold and warm phases; there is an average 7 °C difference in temperature between cold and warm phases; a 10% difference in RH; wind speed is twice as high in the warm phase compared to the cold phase; and solar radiation is approximately twice as high in the warm phase than in the cold phase. The appreciable differences in all of the aforementioned parameters significantly influence atmospheric stability and mixing height in different phases, favoring the wind-induced resuspension of road dust and crustal material (i.e. soil), thereby enhancing the contribution of soil/road dust in the warm phase compared to the cold phase. Hasheminassab et al. (2014) also found in a PMF source apportionment study across the state of California that the "soil" factor had a maximum contribution in summer (due to the increased temperature and wind speed, as well as atmospheric instability), particularly in areas with lower levels of RH and precipitation, which is consistent with the results observed in the present study.

**(E) Some recent papers have attributed a significant proportion of particulate matter mass in California to emissions from cooking. Such attribution usually comes from AMS data, largely on the basis of diurnal profiles. Is there any indication of such a source within this data set? One might expect it to be more prominent at the weekend than on weekdays, especially if related to outdoor barbequing.**

**Response:** The reviewer's comment is properly taken and the response is provided in detail below.

Although we acknowledge the existence of "cooking" source in the study area, we did not expect that the PMF would be able to resolve such a factor. This is mainly because our sampling site is located in the vicinity of a major freeway (about 150 m downwind), in the middle of a parking center, and is quite far from (at least obvious) sources of food cooking emissions (e.g. restaurants, residential areas). So we can conclude that compared to traffic sources, the contribution of such a minor source, such as cooking, would be very small, making it difficult to be detected/resolved by the PMF model. Moreover, previous studies conducted at the same sampling location were not able to identify/quantify "cooking" sources even using specific tracers of cooking, such as cholesterol, in the chemical mass balance (CMB) source apportionment model (Arhami et al., 2010;Hasheminassab et al., 2013).

Nonetheless, the contribution from "cooking" sources, although very minor, is expected to be within the resolved ""urban background aerosol" factor. However, since many sources are involved in this factor to form the background concentrations, detecting a distinct diurnal trend from each of the participating sources is quite unlikely.

**(F) Page 3, line 6. This implies that regulations on PM number emissions from road vehicles in Europe have been set on the basis of health studies of UFPs. This is not the case. The PMP number limits, which have been introduced as part of recent European vehicle emission standards, arose because of the difficulty encountered in achieving repeatable measurements of very low mass concentrations of PM in diluted engine exhaust, whereas particle number could be measured very repeatedly after removable of the semi-volatile fraction. Hence a test procedure, based upon solid particle number, was considered a more practicable option than determination of mass.**

**Response**: We thank the reviewer for his/her insightful comment. To address the reviewer's comment we changed the sentence in the introduction in a way to dispel the implication that the European particle number concentration standard was set because of the body of evidence linking health effects to particle number concentrations. The revised paragraph can be found below.

Page 3, Lines 6-7: "Regulations on PM number concentrations have already been implemented on motor vehicle emissions in a few countries."

**G. Page 5, line 34. Last word should read 'factor' rather than 'actor'.**

**Response**: The reviewer's comment was noted and the change was made in the revised manuscript (Page 5, Line 32).

**H. Page 6, line 30. 'Ration' should read 'ratio'.**

**Response**: The reviewer's comment was noted and the change was made in the revised manuscript (Page 6, Line 31).

**I. It would be good to quote the OC/EC gradient determined through equation 1, as there is still considerable interest in applications of the EC Tracer Method to source apportionment of organic carbon.**

**Response**: The reviewer's comment is properly taken and the following information was added to the revised manuscript accordingly.

Page 7, Lines 1-3: "Using equation (1), the slope and the intercept of the regression line were found to be 1.55 (±0.07) and 0.45 (±0.24), respectively. More detailed information on the results obtained using the EC tracer method can found elsewhere (Saffari et al., 2016)"

**References**

Arhami, M., Minguillón, M. C., Polidori, A., Schauer, J. J., Delfino, R. J., and Sioutas, C.: Organic compound characterization and source apportionment of indoor and outdoor quasi‐ultrafine particulate matter in retirement homes of the Los Angeles Basin, Indoor Air, 20, 17-30, 2010.
Beddows, D. C. S., Harrison, R. M., Green, D. C., and Fuller, G. W.: Receptor modelling of both particle composition and size distribution from a background site in London, UK, Atmospheric Chemistry and Physics Discussions, 15, 10123-10162, 2015.

Charron, A., and Harrison, R. M.: Primary particle formation from vehicle emissions during exhaust dilution in the roadside atmosphere, Atmospheric Environment, 37, 4109-4119, 2003.

Friend, A. J., Ayoko, G. A., Jayaratne, E. R., Jamriska, M., Hopke, P. K., and Morawska, L.: Source apportionment of ultrafine and fine particle concentrations in Brisbane, Australia, Environmental Science and Pollution Research, 19, 2942-2950, 2012.

Harrison, R. M., Beddows, D. C. S., and Dall'Osto, M.: PMF analysis of wide-range particle size spectra collected on a major highway, Environmental science & technology, 45, 5522-5528, 2011.

Hasheminassab, S., Daher, N., Schauer, J. J., and Sioutas, C.: Source apportionment and organic compound characterization of ambient ultrafine particulate matter (PM) in the Los Angeles Basin, Atmospheric Environment, 79, 529-539, 2013.

Janhäll, S., Jonsson, Å. M., Molnár, P., Svensson, E. A., and Hallquist, M.: Size resolved traffic emission factors of submicrometer particles, Atmospheric Environment, 38, 4331-4340, 2004.

Krecl, P., Hedberg Larsson, E., Ström, J., and Johansson, C.: Contribution of residential wood combustion and other sources to hourly winter aerosol in Northern Sweden determined by positive matrix factorization, Atmospheric Chemistry and Physics, 8, 3639-3653, 2008.

Ntziachristos, L., Ning, Z., Geller, M. D., and Sioutas, C.: Particle concentration and characteristics near a major freeway with heavy-duty diesel traffic, Environmental science & technology, 41, 2223-2230, 2007.

Ogulei, D., Hopke, P. K., and Wallace, L. A.: Analysis of indoor particle size distributions in an occupied townhouse using positive matrix factorization, Indoor Air, 16, 204-215, 2006a.

Ogulei, D., Hopke, P. K., Zhou, L., Pancras, J. P., Nair, N., and Ondov, J. M.: Source apportionment of Baltimore aerosol from combined size distribution and chemical composition data, Atmospheric Environment, 40, 396-410, 2006b.

Ogulei, D., Hopke, P. K., Chalupa, D. C., and Utell, M. J.: Modeling source contributions to submicron particle number concentrations measured in Rochester, New York, Aerosol Science and Technology, 41, 179-201, 2007.

Saffari, A., Hasheminassab, S., Shafer, M. M., Schauer, J. J., Chatila, T. A., and Sioutas, C.: Nighttime Formation of Aqueous-Phase Secondary Organic Aerosols in Los Angeles and its Implication for Fine Particulate Matter Composition and Oxidative Potential., Atmospheric Environment, under review, 2016.

---

## Author Comment (AC2) · 11 Mar 2016

**Reviewer #2:**

**The study is focused on the characterization of the major sources of PM number concentrations and on the quantification of their contributions using the PMF receptor model applied to PM number size distributions combined with several auxiliary variables in central Los Angeles. The topic is interesting, the data set is large and reliable, the paper is well organized and data interpretation seems to be sound. While there are many articles regarding aerosol source identification by PMF, there are few regarding the analysis of particle number concentrations by PMF. This work gives a very good quantitative identification of the sources which contribute to the particle number as a function of particle size in Los Angeles. I have only few minor remarks:**

**Authors:** We thank the reviewer for his/her valuable comments on the paper that has improved the quality of the work. Please find below our detailed response and the modification made to the manuscript according to each comment.

**1) p 14 l.32-34: why the same explanation is not valid for traffic 1?**

**Response**: The reviewer's comment is properly noted and the response is provided in details below.

As mentioned in the manuscript, because of its larger mode diameter, "Traffic 2" factor was attributed to more aged particles that could also have come from more distant sources, compared to the "Traffic 1" factor that is attributed to freshly emitted particles from nearby traffic sources. Therefore, one possible source that could affect Traffic 2, but not Traffic 1, is the increased traffic volume in Downtown Los Angeles (located 4 km northeast of our sampling site), mainly due to the increased nighttime activities in this part of the city, especially during weekend nights. To further support this hypothesis, we analyzed the wind direction data during our sampling campaign and realized that in weekends, the prevailing wind was from NE direction during night, as can be seen in the table below. This observation further corroborates our hypothesis that Traffic 2 particles

could have come from Downtown LA during weekend nights, when the nighttime activities, and therefore traffic volume, peak in this area of the city. These sources are sufficiently far not to affect the Traffic 1 factor (i.e. freshly emitted particles), because by the time these particles reach our sampling site, their size would have grown to larger ranges not captured in Traffic 1 factor.

Average wind speed and wind direction during the sampling campaign in central Los Angeles

| Hour | Wind speed | | |
|------|------|------|------|
| | Avg | STD | WD |
| 0 | 3.43 | 1.74 | NE |
| 1 | 3.51 | 1.83 | NE |
| 2 | 3.65 | 2.10 | NE |
| 3 | 3.76 | 1.89 | NE |
| 4 | 3.63 | 1.99 | NE |
| 5 | 3.78 | 1.93 | NE |
| 6 | 4.00 | 2.23 | NE |
| 7 | 3.93 | 2.29 | NE |
| 8 | 3.60 | 2.14 | NE |
| 9 | 3.10 | 1.88 | E |
| 10 | 3.36 | 1.81 | S |
| 11 | 3.89 | 1.94 | SW |
| 12 | 5.16 | 2.47 | SW |
| 13 | 6.31 | 2.60 | SW |
| 14 | 7.34 | 2.49 | SW |
| 15 | 7.54 | 2.17 | W |
| 16 | 7.23 | 2.11 | W |
| 17 | 6.14 | 2.19 | W |
| 18 | 4.63 | 2.15 | W |
| 19 | 3.91 | 1.84 | W |
| 20 | 3.45 | 1.70 | W |
| 21 | 3.21 | 1.59 | N |
| 22 | 3.08 | 1.77 | NE |
| 23 | 3.25 | 1.75 | NE |

**2) Factor 5: is there any explanation why this factor gives such a small contribution to particle number? This factor is attributed to secondary nitrates and organics (quite reasonable). Is not there any contribution from secondary sulfates in Los Angeles? If yes, in which of the identified factors is?**

**Response**: The reviewer's comment is properly taken and detailed response is provided below.

Regarding the first part of the comment (i.e., relatively low contribution of this factor to particle number), it should be noted that it is mostly because of the size range in which this factor lies, making this source factor a relatively small contributor to particle number concentrations; as the particles grow in size, their contribution in mass concentrations increases drastically, while their contribution to number concentrations decreases (Figure 4). The "secondary aerosol" source factor identified in this study ranged between 400-500 μm, which, based on the results presented in Figure 2 (i.e. number and volume size distribution) contributes greatly to particle volume/mass concentration, but does not have a large contribution to the particle number concentration. This observation is also consistent with the results presented in many previous studies. For instance, in a study conducted by (Ogulei et al., 2006), the contribution of secondary sources to the total particle number concentrations in Baltimore was found to be in the order of 100 particles/cm3, while the contribution from traffic and other important sources (such as power plant) were found to be in the order of thousands of particles/cm3. (Beddows et al., 2015) also reported that, in London, the contribution of the identified "secondary aerosol" sources was around 400 particles/cm3, while that of traffic sources was as high as 3500 particles/cm3. Additionally, in a study performed by (Friend et al., 2013), a source that was identified as "secondary aerosol" had a contribution to particle number concentration of around 5% in the sampling site that was mostly affected by traffic sources, while its contribution was 10% in the sampling site that was less affected by traffic sources (mainly because the particle number concentrations in the first traffic site were mostly affected by traffic sources, making the relative contribution from other sources less important). This was also the case in the present study, in which traffic sources together

were found to contribute more than 60% to the total particle number concentrations, overwhelming the relative contribution from other sources.

Regarding the second part of the comment (i.e., the possible contribution from secondary sulfate), we do acknowledge the existence of secondary sulfate in Los Angeles, given that this source has been identified in several previous source apportionment studies, using PMF, in this area (Hasheminassab et al., 2014;Kim and Hopke, 2007). However, it should be noted that these studies were performed on particle mass concentrations using chemically-speciated data. Using a chemically-speciated dataset, one can easily distinguish secondary nitrate from secondary sulfate, because of the existence of chemical markers of such sources, i.e. $NO_3^-$ and $SO_4^{2-}$. However, using solely particle number size distribution data, discerning these two factors (along with secondary organic aerosols) is rather impossible, because they lie within the same size range, which is the most important criterion for PMF to differentiate source factors.

It should be noted that the only PMF source apportionment studies on particle number concentrations that have been able to separately identify secondary sulfate and secondary nitrate are those that also included chemically-speciated data along with the particle number size distributions.

Lastly, although the PMF model in this study was not able to identify separate source factors for secondary sulfate and nitrate, the contribution of secondary sulfate is expected to coexist in the "secondary aerosol" factor resolved by the PMF solution. However, we were not able to observe any fingerprints (distinct patterns of diurnal or seasonal variation) pertinent to secondary sulfate. This is mainly because the seasonal and diurnal variations of secondary sulfate and nitrate are actually reverse, the former peaking in summer and in mid-day, while the latter peaking in the cold season and at night. Moreover, previous studied in this area (Hasheminassab et al., 2014) found that the concentrations of secondary nitrate aerosols far exceed those of secondary sulfate (2-3 times higher); therefore, it can be concluded that the seasonal as well as the diurnal variation of the "secondary aerosol" resolved in this study are probably governed by the contributions of secondary nitrate aerosols, as these particles are believed to be the major component of the secondary aerosols, at least in the Los Angeles area.

**3) p 18 l.9: the factor was called "soil/road dust" but the time trend of this source does not justify road dust as a source**

**Response**: The reviewer's comment is properly noted and a detailed response is provided below.

Based on the previous studies conducted in Los Angeles at the same sampling location (Cheung et al., 2012;Shirmohammadi et al., 2015), and also given the close proximity of our sampling site to major traffic sources (surface streets and freeways), we are confident that the "road dust" exists as a source of PM in the study area. However, as mentioned above, since we only used particle number size distribution data for the source apportionment analysis without any chemically-speciated data or unique source tracers, the PMF model could not separately identify two distinct "road dust" and "soil" factors. Nonetheless, given the typical size range of road dust particles, which mainly exist in the coarse mode (Cheung et al., 2012), we believe that road dust would be partitioned in the identified "soil" factor as well.

Additionally, based on the study of (Cheung et al., 2012), the contribution of "road dust" is expected to be higher in summer, which is consistent with the seasonal trend observed for the "soil/road dust" factor in this study, with much higher contribution in the warm phase (Figure 6).. Moreover, prior studies in the same area indicated that the contribution of soil is larger compared to road dust (Cheung et al. 2012) Therefore, the overall diurnal trend for this factor, i.e. soil/road dust, may have been dominated by the soil particles, not enabling us to see a distinct diurnal variability between the two sources.

**4) Fig 3: the normalized concentrations of PM2.5-10 are higher respect to PM2.5 for both traffic 2 and urban background; is there any possible explanation?**

**Response**: The reviewer's comment is properly taken and the author's response is provided in details below.

We concur with the reviewer that the size range of particles identified in the "Traffic 2" and "Urban Background" are larger than those of the "Traffic 1" factor (as shown by the number size distributions). This is mainly because that they are more aged and more likely to originate from more distant sources (especially the urban background aerosol) compared to the freshly emitted particles that come from the "Traffic 1" factor. However, this cannot solely justify the relatively high loadings of PM10-2.5 in these two factors. This is due to the fact that PM10-2.5 particles are different in sources and chemical composition compared to PM2.5, and the only factor, among all of the identified source factors, that the can be attributed to this size fraction is "soil/road dust" (Figure 2). Therefore, we believe the loading of PM10-2.5 observed in these two factors (around 20%) may reflect a PMF artifact, which may occasionally fail to distinguish the profiles of two or more sources, and on several instances residuals from other factors may be observed in the factor of interest, as also noted by reviewer #1.

Additionally, it should be noted that our main scope in this study was particle number apportionment, rather than particle mass apportionment. Mass concentrations, along with some other parameters (e.g. gaseous pollutants, traffic data, etc.) were included as auxiliary data only to help the interpretation of the resolved factors. A sensitivity analysis was run to identify the impact of these auxiliary variables on the PMF results. As can be seen below, results of the analysis indicated that the results of the PMF model are quite robust even after excluding the data pertaining to the auxiliary variables (i.e., PM mass, gaseous pollutants, EC/OC, meteorological parameters, and traffic count data).

[Figure]

Base Factor Profiles

**5) Fig. 8: why on weekend there is a night peak only for traffic 2?**

**Response:** We have fully addressed this comment in response to the first comment raised by the respected reviewer. So you are kindly referred to the response to the first comment.

**References**

Beddows, D. C. S., Harrison, R. M., Green, D. C., and Fuller, G. W.: Receptor modelling of both particle composition and size distribution from a background site in London, UK, Atmospheric Chemistry and Physics Discussions, 15, 10123-10162, 2015.

Cheung, K., Shafer, M. M., Schauer, J. J., and Sioutas, C.: Diurnal trends in oxidative potential of coarse particulate matter in the Los Angeles Basin and their relation to sources and chemical composition, Environmental science & technology, 46, 3779-3787, 2012.

Friend, A. J., Ayoko, G. A., Jager, D., Wust, M., Jayaratne, E. R., Jamriska, M., and Morawska, L.: Sources of ultrafine particles and chemical species along a traffic corridor: comparison of the results from two receptor models, Environmental Chemistry, 10, 54-63, 2013.

Hasheminassab, S., Daher, N., Saffari, A., Wang, D., Ostro, B. D., and Sioutas, C.: Spatial and temporal variability of sources of ambient fine particulate matter (PM2.5) in California, Atmos. Chem. Phys., 14, 12085-12097, 10.5194/acp-14-12085-2014, 2014.

Kim, E., and Hopke, P. K.: Source characterization of ambient fine particles in the Los Angeles basin, Journal of Environmental Engineering and Science, 6, 343-353, 2007.

Ogulei, D., Hopke, P. K., Zhou, L., Pancras, J. P., Nair, N., and Ondov, J. M.: Source apportionment of Baltimore aerosol from combined size distribution and chemical composition data, Atmospheric Environment, 40, 396-410, 2006.

Shirmohammadi, F., Hasheminassab, S., Wang, D., Saffari, A., Schauer, J. J., Shafer, M. M., Delfino, R. J., and Sioutas, C.: Oxidative potential of coarse particulate matter (PM 10–2.5) and its relation to water solubility and sources of trace elements and metals in the Los Angeles Basin, Environmental Science: Processes & Impacts, 17, 2110-2121, 2015.